# Protection against overfeeding-induced weight gain is preserved in obesity but does not require FGF21 or MC4R

Camilla Lund [1,3], Pablo Ranea-Robles [1,3], Sarah Falk [1], Dylan M. Rausch [1], Grethe Skovbjerg[1,2], Victoria Kamma Vibe-Petersen[1], Nathalie Krauth [1], Jacob Lercke Skytte [2], Vasiliki Vana[1], Urmas Roostalu [2], Tune H. Pers [1], Jens Lund [1] & Christoffer Clemmensen [1] ✉

Overfeeding triggers homeostatic compensatory mechanisms that counteract weight gain. Here, we show that both lean and diet-induced obese (DIO) male mice exhibit a potent and prolonged inhibition of voluntary food intake following overfeeding-induced weight gain. We reveal that FGF21 is dispensable for this defense against weight gain. Targeted proteomics unveiled novel circulating factors linked to overfeeding, including the protease legumain (LGMN). Administration of recombinant LGMN lowers body weight and food intake in DIO mice. The protection against weight gain is also associated with reduced vascularization in the hypothalamus and sustained reductions in the expression of the orexigenic neuropeptide genes, *Npy* and *Agrp*, suggesting a role for hypothalamic signaling in this homeostatic recovery from overfeeding. Overfeeding of melanocortin 4 receptor (MC4R) KO mice shows that these mice can suppress voluntary food intake and counteract the enforced weight gain, although their rate of weight recovery is impaired. Collectively, these findings demonstrate that the defense against overfeeding-induced weight gain remains intact in obesity and involves mechanisms independent of both FGF21 and MC4R.

Body weight and fat mass are under homeostatic regulation, and feedback mechanisms affecting energy intake and energy expenditure are evoked to counter perturbations in energy balance. This aligns with the 'dual intervention point' model of body weight regulation, which argues that body weight is maintained between an upper and a lower boundary (so-called biological intervention points), rather than being regulated around a specific 'set point'[1–5]. This homeostatic system may have evolved to help ensure mammalian survival by lowering the risk of predation (carrying too much fat tissue) and starvation/sickness-induced anorexia (having too little fat mass)[6]. Most remarkable is the efficiency by which the organism counteracts conscious attempts to lower body weight by increasing appetite and reducing energy

expenditure, rendering dieting and other lifestyle interventions futile for sustained weight loss in many individuals[7–9]. In the context of a chronic negative energy balance, reduced levels of the adipocyte-derived hormone leptin are acknowledged as a primary molecular driver of weight regain[10–12].

Importantly, the homeostatic feedback mechanisms that protect body fat mass from large fluctuations are also triggered by conscious attempts to gain weight. This is illustrated by numerous overfeeding studies in both animals[13] and humans[14] in which a controlled weight gain is recovered after overfeeding is stopped. However, the molecular mediators of this biological defense remain unknown[15,16]. Moreover, it is unclear whether obesity influences the homeostatic recovery from

[1]Novo Nordisk Foundation Center for Basic Metabolic Research, University of Copenhagen, Copenhagen, Denmark. [2]Gubra ApS, Hørsholm, Denmark. [3]These authors contributed equally: Camilla Lund, Pablo Ranea-Robles. ✉e-mail: chc@sund.ku.dk

overfeeding-induced weight gain. Parabiosis studies in rodents reveal that obesity induction in one partner can lower food intake and fat mass in the other, suggesting endocrine regulation[17–19]. However, leptin's negligible role in countering chronic overfeeding in mice[20] highlights the involvement of yet-to-be-identified signaling molecules and regulatory pathways in the physiological protection against weight gain[15,16].

In this work, we employed an intragastric overfeeding mouse model to investigate the role of obesity in the homeostatic defense against experimental weight gain. Furthermore, we investigated the causal implication of fibroblast growth factor 21 (FGF21) and the melanocortin 4 receptor (MC4R) in response to overfeeding, and we determined transcriptional and vascular changes in the hypothalamus during this intervention. Finally, we report here the first strides toward

testing the pharmacological effects of potential endocrine regulators of the response to overfeeding.

## Results

### Lean and obese mice are protected against overfeeding-induced weight gain

Intragastric experimental overfeeding (ExpOF), i.e. infusion of a hypercaloric liquid diet to achieve 50% energetic surplus over baseline requirements of lean C57BL/6 J male mice (Fig. 1a, Supplementary Fig 1a, Supplementary Data 1) resulted in an average body weight gain of ~28% over a 14-day period (Fig. 1b). This corresponds to an absolute weight gain of ~8 g (Fig. 1c) with an inter-individual variability of 2.2-fold between the lowest and the highest gainers (Supplementary Fig. 1b). After cessation of overfeeding on day 14 (d14), overfed mice

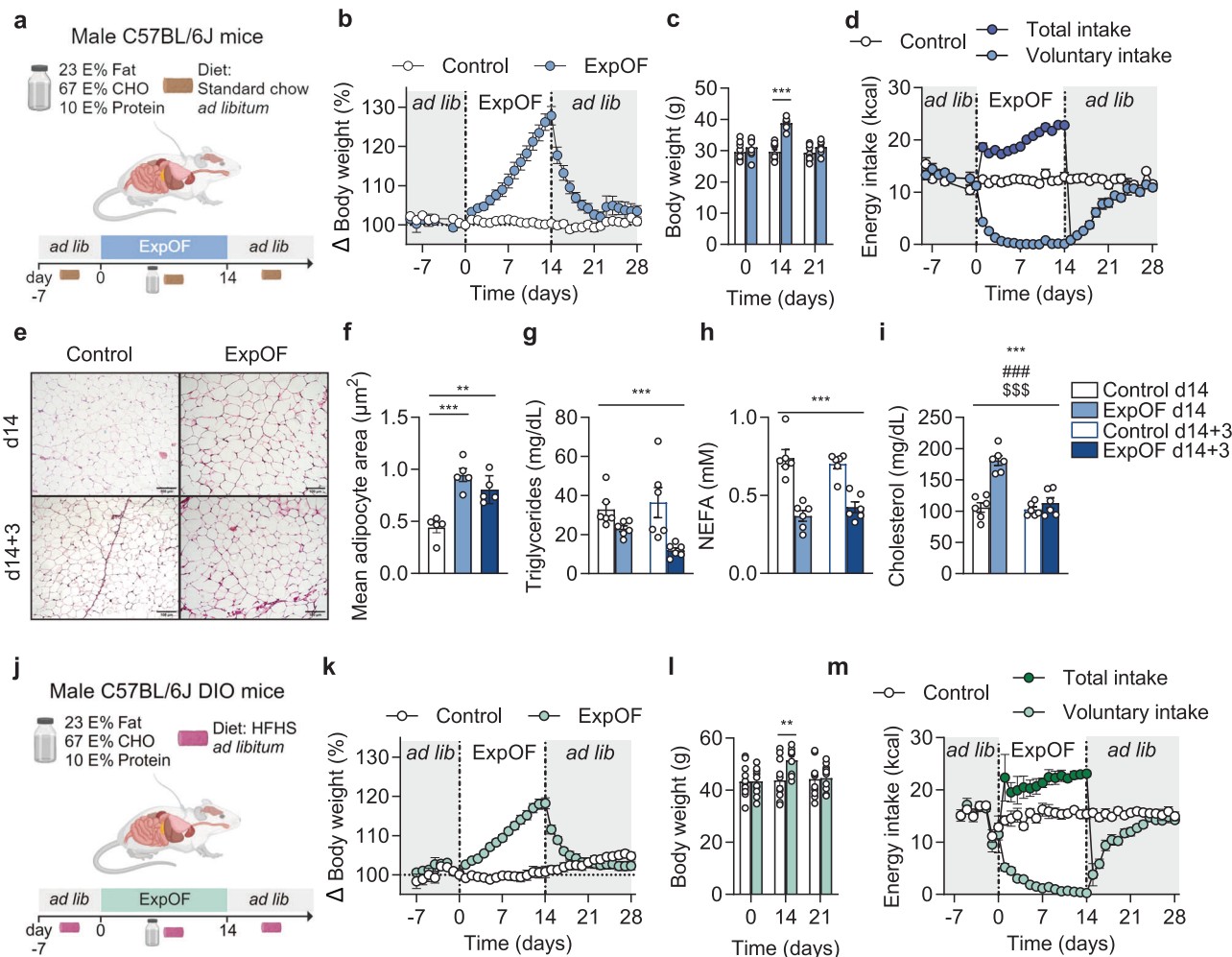

**Fig. 1 | Effects of overfeeding in lean and diet-induced obese mice. a** Schematic overview of the experimental overfeeding (ExpOF) setup in chow-fed lean mice. Created with BioRender.com. **b** Body weight changes (percentage over baseline) in control (*n* = 12) and ExpOF (*n* = 8) mice. Body weight is set at 100% at d0 (start of ExpOF). **c** Absolute body weight (in grams) of the same mice shown in **b** on day 0, 14, and 21. **d** Daily total (dark blue) and voluntary (light blue) energy intake (in kcal) of the same mice shown in **b**. **e** Representative images of H&E-stained eWAT sections of control and ExpOF mice on d14 and d14 + 3 (*n* = 5), quantified in **f**. Scale bar = 100 µm. **f** Quantitative analysis of adipocyte size in eWAT sections from **e**. **g** Plasma triglycerides (TG + free glycerol) levels (in mg/dL) in control and ExpOF mice on d14 and d14 + 3 (*n* = 6). **h** Plasma NEFA levels (in mM) in control and ExpOF mice on d14 and d14 + 3 (*n* = 6). **i** Plasma total cholesterol levels (in mg/dL) in control and ExpOF mice on d14 and d14 + 3 (*n* = 6). **j** Schematic overview of the

ExpOF setup in diet-induced obese (DIO) mice. Created with BioRender.com. **k** Body weight changes (percentage over baseline) in control (*n* = 12) and ExpOF (*n* = 11) DIO mice. Body weight is set at 100% at d0 (start of ExpOF). **l** Absolute body weight (in grams) of same mice shown in **k** on days 0, 14, and 21. **m** Daily total (dark green) and voluntary (light green) energy intake (in kcal) of same mice shown in **k**. Data shown as mean ± SEM with individual values plotted in **c, f–i, l**. *P* values were calculated using 1-way ANOVA and Tukey´s post hoc analysis (**f**), 2-way ANOVA (mixed-effects analysis) using overfeeding and time as factors (**c, g–i, l**). */**/*** was used when *p* < 0.05, *p* < 0.01, *p* < 0.001, respectively, in the post hoc comparisons after ANOVA (**c, f, l**). */#/$; **/##/$$; ***/###/$$$ were used when *p* < 0.05; *p* < 0.01; *p* < 0.001, for overfeeding/time/interaction effects, respectively (**g–i**). *n*: number of mice (biological replicates). Source data are provided as a Source Data file.

recovered their baseline body weight within 7 days (Fig. 1b, c). Voluntary chow intake was completely suppressed throughout the overfeeding period, from day 5 onward (Fig. 1d). This hypophagic response persisted for several days after overfeeding and gradually increased until reaching baseline levels 7 days after overfeeding (Fig. 1d). No changes in body weight or voluntary food intake were observed in control mice infused with equal volumes of water (Fig. 1b–d). Despite increased energy efficiency (mg of weight gain per kcal consumed) during overfeeding compared to controls (Supplementary Fig. 1c), overfed mice consumed more calories than control mice throughout the entire study (Supplementary Fig. 1d), suggesting that energy expenditure and/or energy excretion may play differential roles in the distinct phases of overfeeding and recovery[21,22]. These findings align with previous rodent studies[13,20,23–26], highlighting that appetite suppression plays a crucial role in the defense against overfeeding-induced weight gain. Further research is warranted to elucidate the potential involvement of energy expenditure and energy excretion in experimental overfeeding.

Overfeeding increased adipocyte size in epididymal white adipose tissue (eWAT) (Fig. 1e, f) and increased hepatic lipid content (Supplementary Fig. 1e). These changes were sustained for 3 days into the recovery period (d14 + 3) (Fig. 1e, f and Supplementary Fig. 1e). Morphological changes in eWAT and liver coincided with transcriptional changes related to fuel substrate metabolism and adipose extracellular matrix remodeling (Supplementary Fig. 1f, h, i, k). In contrast, no evidence of endoplasmic reticulum (ER) stress or inflammation was observed (Supplementary Fig. 1g, j). This latter observation contrasts with high-fat diet-induced obesity (DIO) in mice, which induces inflammation and ER stress, but is consistent with a previous overfeeding study[20]. Plasma triglyceride levels were reduced by ~50% on day 3 of the recovery period (Fig. 1g). In addition, plasma non-esterified fatty acids (NEFA) levels were decreased by ~50% on d14, and this effect was sustained for 3 days into the recovery period (Fig. 1h). In contrast, total cholesterol was increased by ~80% after overfeeding but was normalized 3 days into the recovery period (Fig. 1i). The decrease in plasma NEFA is also evident in human overfeeding studies[14], and might reflect preserved insulin sensitivity and thus an increased efficiency to inhibit lipolysis during the diet infusion.

To investigate whether obesity alters the homeostatic defense against overfeeding, we subjected DIO mice to 14 days of intragastric overfeeding with ad libitum access to high-fat diet (Fig. 1j, Supplementary Data 1). DIO mice gained on average ~18% body weight during overfeeding (Fig. 1k), corresponding to ~8 g of absolute weight gain (Fig. 1l), as seen in overfed lean mice (Fig. 1c). Overfed DIO mice returned to baseline body weight 6 days after overfeeding (Fig. 1k), comparable to the duration of weight recovery observed in lean mice (Fig. 1b). Voluntary intake of high-fat diet gradually decreased in DIO mice during overfeeding and was almost completely suppressed by day 10 (Fig. 1m). In contrast, overfed lean mice completely stopped voluntary chow intake after 5 days of overfeeding (Fig. 1d). The delayed suppression of voluntary eating in DIO mice during overfeeding may be related to the increased palatability of the high-fat diet[26] and/or to their elevated baseline body weight. Overfed DIO mice slowly recovered their appetite and reached baseline levels of voluntary food intake 7 days after overfeeding (Fig. 1m), similar to the pattern observed in overfed lean mice (Fig. 1d). Although lean and obese mice appear to exhibit temporal differences in their hypophagic responses to overfeeding, these data demonstrate that the overall physiological defense against overfeeding-induced weight gain remains intact in DIO mice.

## FGF21 is dispensable for the physiological protection against weight gain

Overfeeding increased circulating levels of glucose, insulin, and leptin, peaking on the final day of overfeeding (Fig. 2a–c). Conversely, plasma ghrelin levels were significantly decreased on d14 (Fig. 2d). While glucose and insulin levels returned to baseline within a day after termination of overfeeding (Fig. 2a, b), plasma leptin levels declined more gradually and reached baseline levels after 3–4 days of recovery (Fig. 2c). Accordingly, the correlation between leptin levels and body weight observed during overfeeding was lost 3–4 days into the recovery period (Supplementary Fig. 2a, b). Of note, this normalization of circulating leptin occurred several days before the full restoration of voluntary food intake (Fig. 1d). Ghrelin levels were no longer suppressed 3 days into the recovery period (Fig. 2d). These findings are consistent with previous rodent overfeeding studies demonstrating a temporal uncoupling between the normalization of circulating levels of insulin, leptin, and ghrelin and the extended duration of the hypophagic recovery period post-overfeeding, which persists beyond the normalization of these hormones[20,23,25].

These findings also emphasize that unidentified endocrine signals —different from the classical appetite-regulating hormones—might mediate the potent and prolonged hypophagia observed after overfeeding. FGF21 is a hormone that has been linked to weight gain resistance in humans[27] and several studies have reported that overfeeding increases circulating levels of FGF21[28–30]. Notably, we found a remarkable 10-15-fold induction of Fgf21 expression in white fat, a 15-fold increase in brown fat, and a 30-40-fold increase in the liver (Fig. 2e), which coincided with elevated plasma FGF21 that were 9.5-fold higher in overfed mice compared to controls at d14 (2315 pg/mL vs 243 pg/mL) (Fig. 2f). Interestingly, plasma FGF21 levels further increased to ~5000 pg/mL 3 days into the recovery phase (Fig. 2f). Given this striking increase in circulating FGF21 levels, we directly tested whether FGF21 acts as an endocrine mediator of the homeostatic response to overfeeding by subjecting FGF21 KO mice to intragastric overfeeding (Fig. 2g). FGF21 KO mice gained on average ~26% weight after 14 days of overfeeding (Fig. 2h), which is equivalent to an increase of ~7 grams (Fig. 2i). Similar to what was observed for WT mice (Fig. 1b, c), overfed FGF21 KO mice returned to baseline body weight ~7 days after termination of overfeeding (Fig. 2h, i). Voluntary intake of the chow diet was suppressed during overfeeding and normalized 7-8 days after overfeeding in FGF21 KO mice (Fig. 2j). These findings demonstrate that although FGF21 is abundantly elevated in response to overfeeding, it is dispensable for the homeostatic protection against overfeeding-induced weight gain.

GDF15 is another hormone that has been linked to regulation of energy balance[31]. We observed a pronounced increase of Gdf15 mRNA levels in metabolic tissues after two weeks of overfeeding and during the recovery period, most notably an 80-fold increase in the interscapular brown adipose tissue (iBAT) (Supplementary Fig. 2c). However, circulating GDF15 levels were not changed after 14 days of overfeeding and only slightly elevated during the recovery period (Supplementary Fig. 2d), contrasting with the pronounced transcriptional induction of Gdf15 induced by overfeeding at both time points. The plasma data aligns with human studies, which have failed to identify an increase in circulating GDF15 levels in response to overfeeding[31,32]. Moreover, the plasma levels of GDF15 during and after overfeeding are markedly below the levels known to acutely suppress food intake in mice[33,34].

## Targeted plasma proteomics identify legumain as a potential modulator of the response to overfeeding

Seeking to identify circulating factors of overfeeding, we employed targeted plasma proteomics and identified 6 proteins that were significantly affected by overfeeding: tumor necrosis factor receptor superfamily member 12 A (TNFRSF12A), delta-like protein 1 (DLL1), immunoglobulin superfamily member 3 (IGSF3), tumor necrosis factor receptor superfamily member 27 (EDA2R), legumain (LGMN) and cysteine-rich motor neuron 1 protein (CRIM1) (Fig. 2k, Supplementary Data 2). To gauge the functional role of these circulating factors of

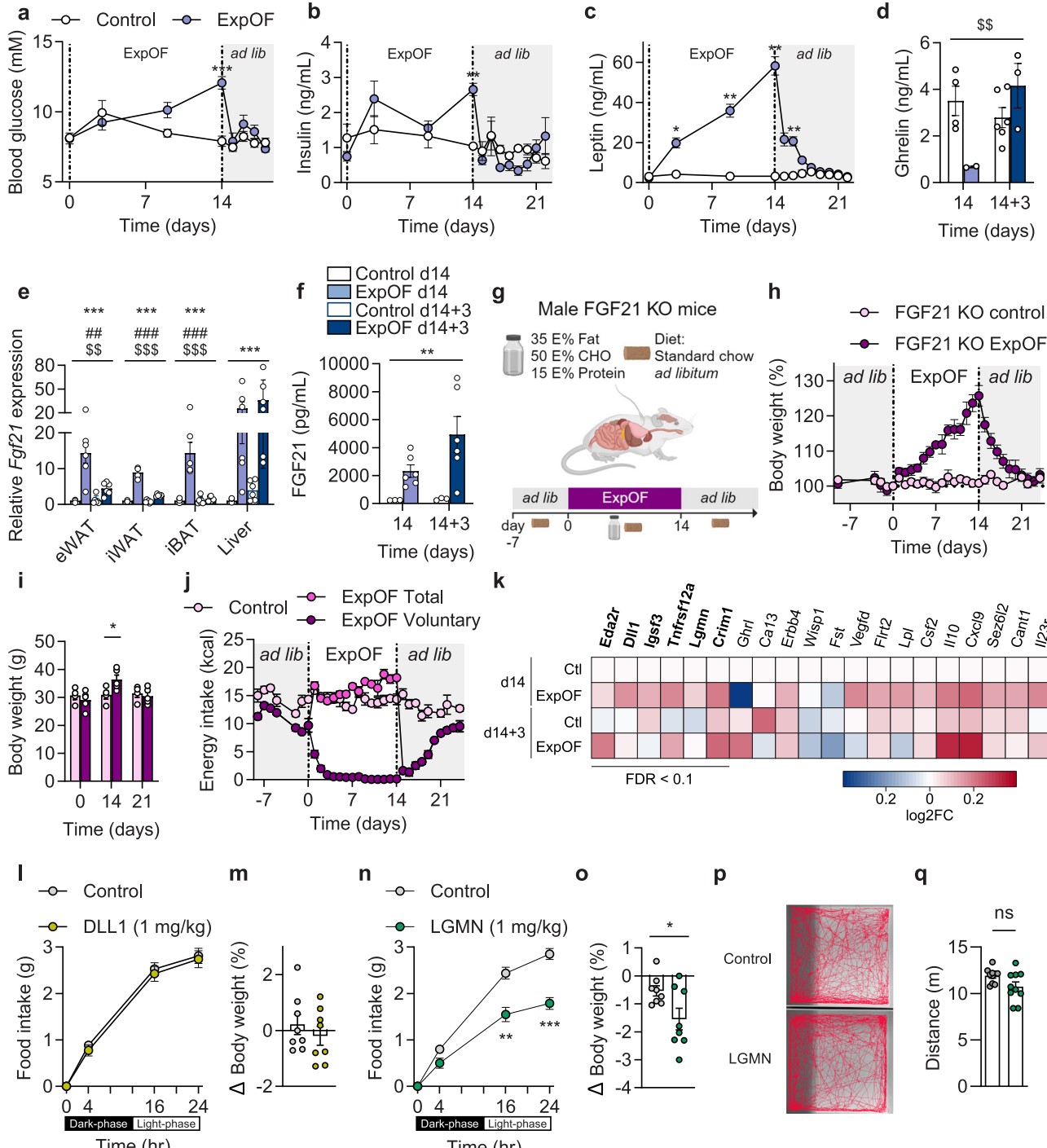

**Fig. 2 | Evaluation of overfeeding-induced changes in potential modulators of energy balance and their role in the response to overfeeding. a–d** Blood glucose (**a**, $n = 5$ per group), plasma insulin (**b**, $n = 5$ per group), plasma leptin (**c**, $n = 5$ per group), and plasma ghrelin (**d**, $n = 4$ Control d14, $n = 6$ Control d14 + 3, $n = 2$ ExpOF d14, $n = 3$ ExpOF d14 + 3) levels at different time points during the overfeeding intervention. **e** *Fgf21* expression in eWAT, iWAT, iBAT, and liver ($n = 6$). **f** Plasma FGF21 levels ($n = 6$). **g** Schematic overview of the ExpOF setup in chow-fed FGF21 KO mice. Created with BioRender.com. **h** Body weight changes (percentage over baseline) in control ($n = 4$) and ExpOF ($n = 5$) FGF21 KO mice. Body weight is set at 100% at d0 (start of ExpOF). **i** Absolute body weight of same mice shown in **h** on day 0, 14, and 21. **j** Daily total (light pink for control mice, dark pink for ExpOF mice) and voluntary (purple for ExpOF mice) energy intake of same mice shown in **h**. **k** Heatmap showing average changes of top 20 regulated proteins (sorted by adjusted $p$ value) between control and ExpOF mice at d14 and d14 + 3 ($n = 6$). Heatmap shows fold-changes in normalized protein expression relative to control d14 mice. Red indicates high relative protein abundance and blue indicates low relative protein abundance. Regulated proteins (FDR < 0.1) are highlighted in bold. **l–o** Effects on food intake (**l**, **n**), and body weight (**m**, **o**) in C57BL/6 J DIO male mice injected once with 1 mg/kg of DLL1 (**l**, **m**) or LGMN (**n**, **o**) recombinant proteins or vehicle ($n = 8$ per group). **p, q** Open field test running traces (**p**) and total distance (**q**) of chow-fed lean C57BL6/J mice injected with LGMN (1 mg/kg) or vehicle ($n = 10$ per group). Data shown as mean ± SEM with individual values plotted (**d–f**, **l**, **m**, **o**, **q**). *P* values were calculated using 2-way ANOVA using overfeeding and time as factors (**a–f**, **h–j**, **l**, **n**), or unpaired two-tailed Welch's *t* test (**m**, **o**, **q**). */**/*** was used when $p < 0.05$, $p < 0.01$, $p < 0.001$, respectively, in Welch's *t* test or post hoc comparison after ANOVA (**a–c**, **i**, **l–o**). */#/$; **/##/$$; ***/###/$$$ were used when $p < 0.05$; $p < 0.01$; $p < 0.001$, for overfeeding/time/interaction effects, respectively (**d–f**). ns: non-significant. $n$: number of mice (biological replicates). Source data are provided as a Source Data file.

overfeeding on energy balance, the two most upregulated non-receptor proteins (DLL1 and LGMN) were generated recombinantly and administered to DIO male mice. Administration of recombinant DLL1 had no acute effects on food intake or body weight in DIO mice (Fig. 2l, m). Conversely, a single injection of LGMN resulted in a marked reduction in voluntary food intake and body weight after 24 hours (Fig. 2n, o). No obvious signs of sickness were observed in the mice when performing the study. A subsequent open field test in lean mice confirmed that LGMN had no distinct acute adverse effects on loco-motor or anxiety-like behavior in the mice (Fig. 2p, q). Whether the pharmacological effects on body weight and food intake also reflect a physiological function of endogenous LGMN should be assessed in future studies.

### Experimental overfeeding has subtle effects on markers of adaptive thermogenesis

While our data indicate that a reduction in food intake is the primary driver of the rapid weight loss following overfeeding in both lean and DIO mice (Fig. 1d, m), the observed alterations in energy efficiency and the total caloric intake (Supplementary Fig. 1c, d) suggest that a concurrent increase in energy expenditure may contribute to the catabolic response upon discontinuation of overfeeding. To probe the potential induction of non-shivering thermogenesis in response to overfeeding, we conducted histological and molecular analyses of adipose tissue and skeletal muscle. Histological analysis of iBAT revealed enlarged lipid droplets in lean mice on d14 (Supplementary Fig. 2e). Overfeeding induced a subtle but persistent increase in *Ucp1* expression in iBAT, lasting for 3 days into the recovery period (Supplementary Fig. 2f). This was accompanied by a 4.5-fold upregulation of *Dio2* expression (Supplementary Fig. 2f), while other thermogenic markers, including *Ppargc1a (Pgc1α)*, *Prdm16*, and *Cidea* were not induced by overfeeding (Supplementary Fig. 2f). In inguinal white adipose tissue (iWAT), we observed a transient 3-fold increase in *Dio2* expression and a 1.5-fold increase in *Prdm16* expression on day 14, but no changes in *Ucp1* and *Pgc1α* mRNA and a decrease in *Cidea* expression at both time points (Supplementary Fig. 2g). Muscle non-shivering thermogenesis markers, including *Ucp3*, *Sln* (sarcolipin), and *Pln* (phospholamban) were downregulated in the quadriceps muscle of overfed lean mice (Supplementary Fig. 2h). These data suggest that adaptive thermogenesis gene programs are activated in brown adipose tissue, but not in muscle, during the hypophagic period in response to overfeeding. However, considering the modest effect sizes, the contribution of overfeeding-induced brown fat thermogenesis appears negligible in comparison to the profound hypophagic response observed during and after overfeeding. Future studies utilizing indirect calorimetry and other measures of heat production in combination with intragastric overfeeding are required to elucidate whether adaptive thermogenesis is involved in the homeostatic defense against overfeeding-induced weight gain.

### Overfeeding triggers transcriptional changes and vascular remodeling in the hypothalamus

To investigate the impact of overfeeding on hypothalamic regulation of energy homeostasis, including the leptin-melanocortin axis, we performed RNA-sequencing on hypothalami from overfed and control wild-type mice on ad libitum chow diet at two time points: after 14 days of overfeeding (d14) and 3 days into the hypophagic recovery period (d14 + 3) (Supplementary Data 3). We detected 165 differentially expressed genes (DEGs) after overfeeding of which 74 were upregulated and 91 were downregulated (Fig. 3a, b). We detected 48 DEGs in the recovery phase between overfed and control mice of which 4 were upregulated and 44 were downregulated (Fig. 3a, c, Supplementary Data 3). The minimal overlap in DEGs between the overfeeding and recovery phases with only 4 shared genes (Fig. 3a) suggests that these periods are characterized by distinct molecular changes in the hypothalamus. This finding supports the notion that the physiological defense against overfeeding-induced weight gain might involve divergent central nervous system (CNS) mechanisms with some being engaged during and others after overfeeding. In addition to the persistent suppression of *Npy* expression observed not only during the 14-day overfeeding period but also extending into the recovery phase (Fig. 3b–d), we identified changes in the expression of other genes involved in the leptin-melanocortin circuit, including a decrease in *Agrp* expression and an increase in *Stat3* expression after 14 days of overfeeding (Fig. 3b, d). Gene Set Enrichment Analysis (GSEA) revealed a positive enrichment score in pathways related to neuronal development, signaling and plasticity in response to overfeeding and during recovery (Fig. 3e, Supplementary Data 4), and a negative enrichment score in pathways related to ribosomal biogenesis and assembly, and mitochondrial oxidative phosphorylation (Fig. 3e). These results indicate that overfeeding induces transcriptional changes in hypothalamic hunger signaling pathways, neuronal plasticity, and mitochondrial bioenergetics.

Hypercaloric diets are reported to elicit cytoarchitectural rearrangements in the hypothalamus, including inflammation and chronic hypervascularization[35,36]. Here, we employed iDISCO 3D imaging, which enables the detection of blood vessel coverage and length relative to the area size of different hypothalamic nuclei: the arcuate nucleus (ARC), the dorsomedial nucleus of the hypothalamus (DMH), and the ventromedial nucleus of hypothalamus (VMH) (Fig. 3f, Supplementary Fig. 3). We observed a small but significant reduction in vessel coverage and length in DMH and VMH of overfed mice at peak of overfeeding and during the recovery phase (Fig. 3g, h, Supplementary Fig. 3). In contrast, no differences in vessel length or coverage were observed in the ARC (Fig. 3i Supplementary Fig. 3).

### MC4R is not required for the defense against experimental weight gain

To evaluate the importance of the hypothalamic melanocortin system in the homeostatic response to overfeeding, we overfed MC4R KO mice while allowing ad libitum access to a chow diet (Fig. 3j, Supplementary Data 1). MC4R KO mice gained an average of ~26% body weight in response to 14 days of overfeeding (Fig. 3k), corresponding to an average absolute gain of ~11 grams (Fig. 3l). This weight gain was comparable to that observed in overfed lean (Fig. 1c) and slightly higher than in DIO mice (Fig. 1l). After overfeeding, MC4R KO mice gradually lost weight, approaching close to baseline levels 11-14 days post-overfeeding (Fig. 3k). This contrasts with lean mice and DIO mice, which had fully returned to baseline body weight within 7 and 6 days, respectively (Fig. 1b, k, Supplementary Fig. 4a). Voluntary chow intake returned close to baseline levels after 6 days in overfed MC4R KO mice (Fig. 3m), slightly earlier than in overfed lean and DIO mice (Fig. 1d, m, Supplementary Fig. 4b). Yet, the voluntary food intake in MC4R KO mice remained just below the level observed in control MC4R KO mice (Fig. 3m). Overfeeding led to elevated plasma leptin levels in MC4R KO mice (Fig. 3n) and a sustained reduction in circulating ghrelin levels that partially persisted for 3 days into the recovery period (Fig. 3o). These findings suggest that while MC4R signaling contributes to the speed of weight loss after experimental overfeeding, it is not essential for the homeostatic recovery of body weight.

## Discussion

In the present study, we established an automated intragastric over-feeding paradigm in mice to interrogate the physiological and molecular mechanisms that protect against overfeeding-induced weight gain. We found that two weeks of overfeeding resulted in a rapid and pronounced weight gain associated with a profound and sustained suppression of voluntary food intake. This hypophagic response is likely the major defense mechanism that effectively counters the increase in body weight. We also show that the homeostatic defense

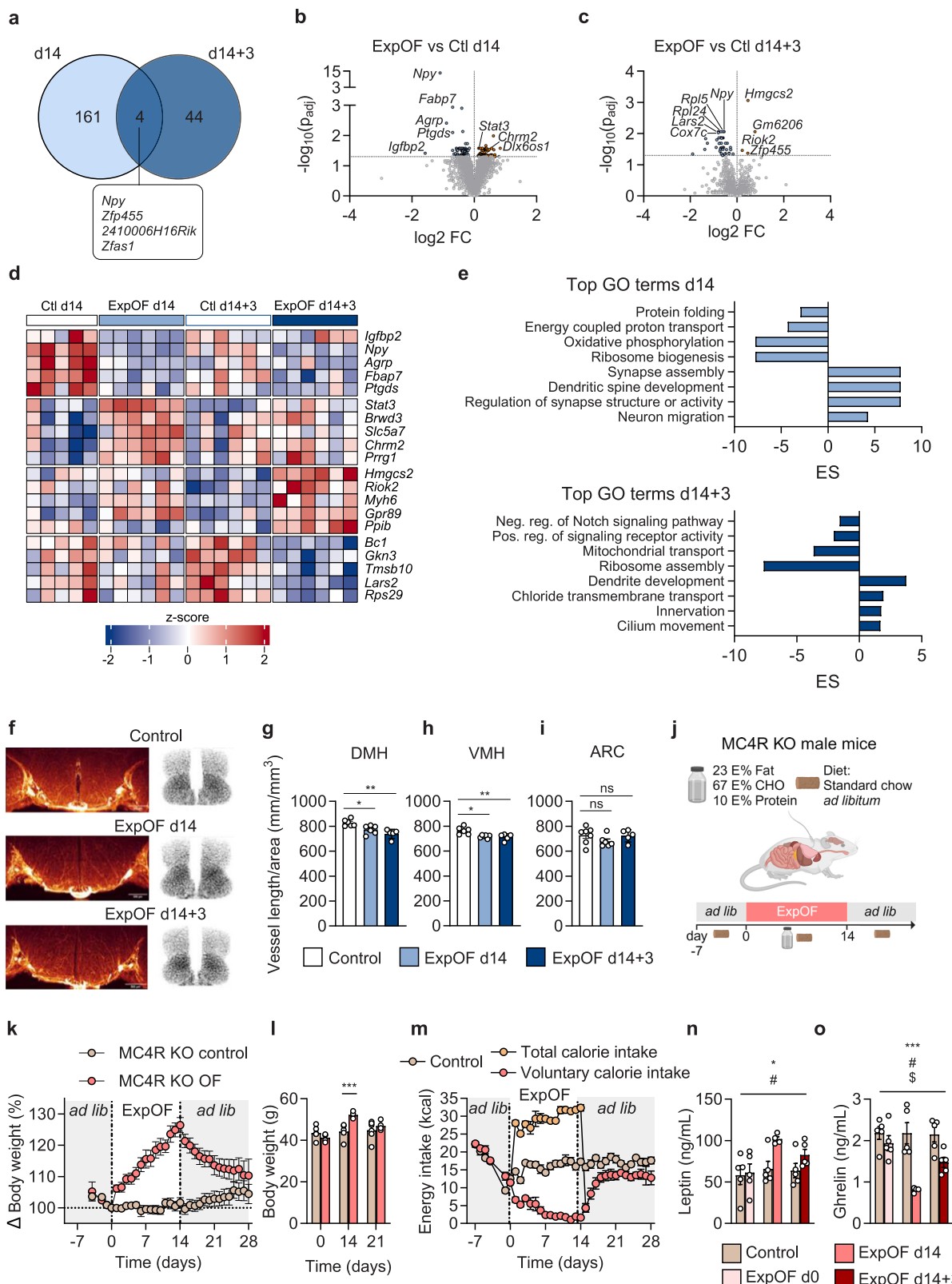

against overfeeding-induced weight gain is unperturbed in high-fat diet-induced obesity and that MC4R deficiency extends the time it takes before body weight is recovered following overfeeding. We demonstrate that FGF21, while markedly elevated in response to overfeeding, is not essential for the homeostatic response against overfeeding-induced weight gain. Finally, we identify circulating proteins linked to overfeeding and take the initial steps to assess their

potential role in body weight regulation, setting the stage for future studies.

High-fat diet (HFD)-induced obesity in male C57BL6 mice is the most established preclinical obesity model[13,37]. Over the course of 3–5 months of HFD-feeding (typically 45–60% energy from fat), mice develop severe obesity coinciding with dyslipidemia and insulin resistance[37]. Resistance to the anorexic effects of exogenous leptin is

**Fig. 3 | Hypothalamic signaling and role of MC4R in overfeeding. a** Venn diagram depicting differentially expressed genes (DEGs) in the hypothalamus of ExpOF mice between d14 and d14 + 3 (Control d14 $n = 5$, ExpOF d14 $n = 6$, Control d14 + 3 $n = 6$, ExpOF d14 + 3 $n = 6$). **b, c** Volcano plots displaying the log$_2$ fold change (FC) vs. −log$_{10}$ of the adj $P$ value ($P_{adj}$) for all genes in the comparison ExpOF vs Control on d14 in **b** and on d14 + 3 in **c**. Contrasts were made with DEseq2 using Wald test and adjusted for multiple comparisons. Significant DEGs are represented with closed blue circles for downregulation and orange circles for upregulation. Selected genes are highlighted. **d** Heatmap showing expression of selected DEGs in the hypothalamus on d14 and d14 + 3. **e** Gene set enrichment analysis of GO biological pathways in ExpOF vs control mice on d14 and d14 + 3. ES enrichment score. **f** Representative iDISCO 3D images of high-resolution hypothalamic areas and schematic visualization of analyzed areas. **g–i** Quantitative measurement of vessel length relative to area in DMH (**g**), VMH (**h**), and ARC (**i**) of control ($n = 7$), ExpOF

d14 ($n = 6$) and ExpOF d14 + 3 ($n = 5$). **j** Schematic overview of the ExpOF setup in chow-fed MC4R KO mice. Created with BioRender.com. **k** Body weight changes during ExpOF of control ($n = 6$) and ExpOF ($n = 5$) MC4R KO mice. **l** Absolute body weight of same MC4R KO mice shown in **k** on day 0, 14 and 21. **m** Daily voluntary energy intake of same MC4R KO mice shown in **k**. Total daily energy intake of ExpOF MC4R KO mice is shown with orange dots. **n, o** Plasma leptin (**n**) and plasma ghrelin (**o**) levels at baseline, on d14 and in the recovery phase in control and ExpOF MC4R KO mice ($n = 5$ in controls, $n = 5$ in ExpOF except at d14 where $n = 4$). Data shown as mean ± SEM with individual values plotted (**g–i, l, n, o**). $P$ values were calculated using 1-way ANOVA (**g–i**), or 2-way ANOVA using overfeeding and time as factors (**l, n, o**). */#/$; **/##/$$; ***/###/$$$ were used when $p < 0.05$; $p < 0.01$; $p < 0.001$, for overfeeding/time/interaction effects, respectively (**n, o**), or */**/*** for individual post hoc comparisons after ANOVA. $n$: number of mice (biological replicates). Source data are provided as a Source Data file.

another hallmark of DIO in mice, and this has been associated with a series of cellular and molecular abnormalities in the hypothalamus, including ER stress[38], inflammation[36], and hypervascularization[39]. Irrespective of the pervasive metabolic abnormalities and disturbances in hypothalamic signaling induced by prolonged high-fat feeding, we demonstrate here that DIO mice efficiently defend themselves against overfeeding-induced weight gain. This is surprising given the aforementioned abnormalities in regulatory mechanisms of energy homeostasis but emphasizes that the protection against experimental weight gain is intact in obese animals.

Although frequently used interchangeably in the literature[13], substantial distinctions exist between HFD-induced obesity and intragastric overfeeding. DIO emerges slowly and does not seem to be effectively counteracted by weight gain defense mechanisms. In contrast, intragastric overfeeding enables an energetic surplus beyond the animal's voluntary consumption, even of the most palatable diet[40]. Notably, whereas voluntary HFD feeding appears to devalue low-fat chow diet[41], intragastric overfeeding of a high-fat liquid diet seems to increase the intake of low-palatable food[26]. Further research is warranted to understand whether intragastric infusion of excess calories through a hypercaloric liquid diet induces a devaluation of chow, similar to the effect observed in mice exposed to HFD. Previous studies have shown that long durations of high-fat diet feeding can lead to persistent metabolic and physiologic changes, even after the obesogenic diet is removed[42]. Only a few studies have explored persistent effects after experimental overfeeding ceases, for example, on adipose tissue remodeling[13]. Despite technical challenges linked to prolonged intragastric overfeeding, future research should consider conducting long-term comparisons between HFD-induced obesity and overfeeding-induced obesity. Furthermore, investigating the impact of varying macronutrient compositions in the overfeeding diet could provide valuable insights into how different dietary components influence weight gain defense mechanisms.

We establish FGF21 as a prominent marker of mouse overfeeding, demonstrating its robust induction in liver, white and brown adipose tissues, and plasma. Previous work has indicated a role for FGF21 in overfeeding[27,29,30,43], and encouraged by the sustained elevation of circulating FGF21 during the recovery period after overfeeding; we functionally evaluated the importance of FGF21 in the homeostatic defense against experimental overfeeding. However, we found that FGF21 KO mice subjected to intragastric overfeeding exhibited weight gain and recovery patterns similar to those of wild-type mice. These observations align with the notion that unidentified endocrine factors mediate the anorectic response to overfeeding[15]. Using a targeted mouse proteomics panel evaluating changes in 92 proteins in response to overfeeding, we identified legumain (LGMN) as an upregulated plasma protein in response to overfeeding in mice. We demonstrate that a single subcutaneous injection of LGMN lowers body weight and food intake in DIO mice. Being a protease, LGMN might exert these effects by cleaving and/or modulating the activity of other proteins

involved in energy homeostasis. Alternatively, LGMN might influence body weight regulation by modulating the reabsorption and lysosomal digestion of macromolecules in the kidney, potentially affecting energy balance[44]. Further research is needed to elucidate the precise role of LGMN in body weight regulation and obesity pathogenesis. However, the potential pharmacotherapeutic exploitation of LGMN for weight loss is discouraged due to its involvement in mechanisms underlying cancer and neurodegenerative diseases[45,46]. As we continue to expand the scope of plasma proteomics in the context of overfeeding, it is crucial to evaluate potential factors using a combination of pharmacological approaches, loss-of-function, and gain-of-function genetic models, and intragastric overfeeding.

The partial involvement of MC4R in weight recovery following overfeeding, along with extensive changes in hypothalamic gene expression and vascularization, strongly indicate that hypothalamic feeding circuits play a crucial role in regulating the upper boundary of weight homeostasis. *Npy* was notably one of the most regulated transcripts, exhibiting suppression both after 14 days of overfeeding and 3 days into the recovery phase. The persistent suppression of *Npy* expression is noteworthy, given the lack of energy infusion and low voluntary food intake during the recovery period. While *Npy* expression typically increases during prolonged fasting[47–49], it paradoxically remains suppressed during the weight recovery period, suggesting the existence of a signal that overrides the usual NPY regulation. Although leptin is known to inhibit hypothalamic *Npy* expression, the normalization of plasma leptin 3 days into the recovery, when *Npy* mRNA is still downregulated, suggests that leptin is not solely responsible for this effect. This observation aligns with previous studies implying that leptin is not an exclusive mediator of the protection against overfeeding-induced weight gain[23,25,50,51], including an overfeeding study using leptin-deficient *ob/ob* mice[20]. In this study, *ob/ob* mice exhibited defense against overfeeding-induced weight gain despite maintaining low and stable blood leptin levels (via mini-pump infusion)[20]. MC4R is a well-established integrator of POMC/CART and AgRP/NPY neuronal activity in feeding regulation. While leptin is a key hormonal regulator of these neuropeptide-expressing hypothalamic neurons, they also respond to other endocrine signals and receive extensive neural input from other brain regions[52]. Thus, overfeeding of MC4R KO mice enabled us to study the necessity of this central receptor within the hypothalamic melanocortin axis for the anorectic response to overfeeding. Although MC4R-deficient mice exhibited a slower weight loss following overfeeding, they ultimately returned close to the body weight of control mice. This suggests that although MC4R somehow modulates the rate of weight recovery, the main mechanism(s) that lowers body weight after overfeeding acts independently of MC4R activation. This finding contrasts with a previous overfeeding study, which found increased *Pomc* expression in the arcuate nucleus of overfed rats and showed a complete abrogation of the post-overfeeding hypophagic response with the administration of an MC3R/MC4R antagonist[53]. Since MC3R is also important for body

weight regulation and for defending against perturbations in energy balance[54,55], additional work is required to parse the relative importance of specific components of the leptin-melanocortin circuit to the homeostatic defense against overfeeding in rodents.

Three-dimensional cytoarchitectural analysis revealed a decrease in hypothalamic vascularization of overfed mice. This finding contrasts with previous studies demonstrating hypothalamic hypervascularization in HFD-induced obesity[35,39]. This discrepancy might reflect differences between voluntary HFD-induced obesity and intragastric overfeeding-induced obesity. Alternatively, subtle temporal differences in study design might explain these dissimilar observations, suggesting a transient decrease in vascularization as an initial response to a hypercaloric diet, potentially reflecting an adaptive mechanism to limit nutrient delivery to specific brain regions.

In conclusion, our findings demonstrate that the homeostatic defense against a controlled weight gain induced by intragastric overfeeding remains intact in obese mice and that the rate by which body weight is recovered after overfeeding is slowed down in MC4R-deficient mice. Furthermore, we demonstrate that FGF21 is dispensable for the anorectic response to overfeeding. Lastly, we show that overfeeding increases circulating levels of LGMN, a protease previously unexplored in the context of obesity. Future studies should investigate whether endogenous LGMN acts as a physiological regulator of energy homeostasis. Our mouse studies unveil that the administration of exogenous LGMN can lower food intake and body weight in DIO mice. This underscores the promise of overfeeding studies as a powerful approach to discovering drug targets for obesity prevention and treatment.

## Methods

### Animals
All mouse studies were conducted at the University of Copenhagen, Denmark, and carried out in accordance with regulations regarding the care and use of experimental animals that were approved by the Danish Animal Experimentation Inspectorate (2018-15-0201-01457). Wild-type (WT) male C57BL/6 J (Janvier, FR) mice, melanocortin 4 receptor (MC4R) knockout mice (The Jackson Laboratory, stock no. 032518, $Mc4r^{tm1Lowl}$), and fibroblast growth factor 21 (FGF21) knockout mice ($Fgf21^{tm1.1Djm}$, The Jackson Laboratory, stock no 033846, available at the CBMR) were used. All experiments were done at 22°C with a 12:12 h light-dark cycle (6am-6pm). Mice had ad libitum access to water and chow diet (SAFE D30, Safe Diets, France, 14% energy from fat, 60% energy from carbohydrates, 26% energy from protein), or when indicated to high-fat high sucrose diet (HFD) (D12331, 58% energy from fat, Research Diets, USA). All mice were single-housed after surgery and during experimental overfeeding and recovery. Mice were euthanized by cervical dislocation, unless trunk blood was collected at the end of the experiment, in which case mice were euthanized by decapitation.

### Experimental overfeeding
**Surgery.** Protocol modified from Ueno et al., 2012 and Ravussin et al. 2018[20,56]. Mice were anaesthetized with 3–4% isoflurane in a gas chamber using SomnoSuite Low-Flow Anesthesia System (Kent Scientific Corporation, USA). Additional systemic (carprofen, 5 mg/kg) and local analgesia were applied on incision sites (lidocaine, 1 mg/mL) before surgery. A longitudinal abdominal incision was made to access the gastric ventricle, and a purse suture was placed above the fundus. The ventricle was punctured in the center of the suture, and an adapted 22ga catheter was inserted in the ventricle (BTPU-040, Instech Laboratories, USA). The suture was tightened and secured, and the catheter was pulled through the abdominal muscle. The abdominal incision was closed with a continuous suture. The catheter end was pulled under the skin to the back of the neck and attached to a 22ga vascular access button (VABM1B/22, Instech Laboratories, USA). The dorsal neck skin incision was closed with a mattress suture. Mice were

placed in their cage over a heating pad until recovery of consciousness. Mice were injected subcutaneously with the analgesic carprofen (5 mg/kg) the next two days after the surgery and allowed to recover for at least five weeks before proceeding with overfeeding.

**Automated overfeeding.** All mice were monitored daily for 1 week prior to being randomized in groups. During this week, daily food intake and body weight were measured to use as baseline data and for calculating energy demands in the overfeeding period. On day −2, 48 hours before starting the overfeeding, mice were connected to infusion pumps (704500, 704501, 703005, 703024, Harvard apparatus, USA) using 22ga tethers with springs (VABM1T/22, Instech Laboratories, USA), polyethylene tubing (BTPE-50, Instech Laboratories, USA), multi-axis lever arms (SMCLA, Instech Laboratories, USA), 22ga swivels (375/22PS, Instech Laboratories, USA) and syringes with luer lock tip (10 mL syringes, 302995; and 20 mL syringes, 302830, Becton and Dickinson, USA). During these 48 hours, the mice were infused with sterile water using the same flow rate as on the first day of overfeeding (Supplementary Data 1).

On day 0, mice in the overfeeding group (ExpOF mice) were infused with a commercial liquid diet (584421, Nutridrink Vanilla, Nutricia, Netherlands) that was supplemented with 20% (w/v) sucrose (S0389, Sigma-Aldrich, USA). The addition of sucrose increased the caloric content of the liquid diet from 1.5 kcal/mL to 2.3 kcal/mL (fats 23 E%, Carbohydrates 67 E%, protein 10E%). To calculate the flow rate of diet infusion, the daily food intake of the chow (3.389 kcal/g) during the baseline period was averaged for all the mice and divided by the caloric density of the diet (2.3 kcal/mL) (macronutrient composition stated in Supplementary Data 1). The flow rate was increased gradually until reaching 150% of energy infusion on day 9 (Supplementary Data 1). The mice were automatically infused and remained connected to pumps 23 hr/day (Supplementary Fig. 1A). Specific flow rates, daily volume infused, and daily caloric infusion are indicated in Supplementary Data 1. Control mice were infused with sterile water using the same flow rates. Syringes were replaced with a fresh liquid diet daily. Mice, water bottles, and chow pellets were weighed every day. Mice were flushed with sterile water for 1 hr (flow: 1 mL/hr) to avoid clogging the tubing and for additional hydration. All tubes and cages were changed on day 7 during overfeeding. On day 14, the overfeeding was terminated, and the gastric infusion was set to 100 μL/hr with sterile water for the overfed mice and control mice, until the mice in the overfeeding group had stabilized their food intake and body weight, to prevent any change in feeding behavior due to stress. Food intake, water intake, and body weights were monitored daily until the end of the study. In case surgery recovery was incomplete, the animals were not included in the overfeeding infusion. If the diet clogged in the internal tubing or if mice got tangled in the spring-tubing system, they were excluded from the study.

**Study 1: Effect of experimental overfeeding on body weight and food intake.** 18-week-old, chow-fed male C57BL6/J mice ($n = 5$ control; $n = 3$ ExpOF) were overfed as described (Supplementary Data 1). Mice were observed after ExpOF to evaluate the changes in body weight and food intake until they stabilized. No tissues were collected following this study.

**Study 2: Effect of experimental overfeeding on circulating factors.** 32-week-old chow-fed male C57BL6/J mice ($n = 5$ control; $n = 5$ ExpOF) overfed as described (Supplementary Data 1). Blood (20 μL) was collected, and glucose (with a glucometer), insulin, and leptin were measured on days 0, 3, 9, and 14 during the overfeeding period, and daily between days 15–22 in the recovery period.

**Study 3: Experimental overfeeding of WT C57BL/6 J mice: tissue collection 1.** 18-week-old chow-fed male C57BL6/J mice ($n = 7$ control;

$n$ = 6 ExpOF) were overfed as described (Supplementary Data 1). On day 14, 3 control and 3 ExpOF mice were sacrificed to collect plasma, eWAT, iWAT, iBAT, liver, muscle, and brain. On day 17 (d14 + 3) 4 control and 3 ExpOF mice were sacrificed, and the same tissues were collected.

**Study 4: Experimental overfeeding of WT C57BL/6 J mice: tissue collection 2.** Study 3 was repeated to supplement tissues for analysis (day 14: $n$ = 3 control, $n$ = 3 ExpOF; day 17: $n$ = 3 control; $n$ = 3 ExpOF). (Supplementary Data 1)

**Study 5: Experimental overfeeding of WT C57BL/6 J mice: collection of brains for iDISCO.** 18-week-old chow-fed male C57BL6/J mice ($n$ = 8 control; $n$ = 11 ExpOF) were overfed as described (Supplementary Data 1). On the day of sacrifice (days 14 and 14 + 3), mice were anesthetized using 4% isoflurane and perfused intracardially for iDISCO staining. The study was performed twice, and groups were pooled. The control groups at d14 and d14 + 3 were pooled (control $n$ = 8, ExpOF d14 $n$ = 6, ExpOF d14 + 3 $n$ = 5).

**Study 6: Experimental overfeeding of DIO C57BL/6 J mice—observation of recovery after overfeeding.** 31-week-old obese male C57BL/6 J mice ($n$ = 12 control; $n$ = 11 ExpOF) that were switched to an HFD (D12331, Research Diets, USA, 58% energy from fat and sucrose, of which 35.4% from fat) at 8 weeks of age were overfed as described (Supplementary Data 1). DIO mice were matched for baseline body weight (control = 43.3 ± 5.7 g, ExpOF = 43.4 ± 4.9 g) and baseline food intake (control = 15.9 ± 1.7 kcal/d, ExpOF = 16.1 ± 1.4 kcal/d) before the start of the infusions.

**Study 7: Experimental overfeeding of MC4R KO mice—observation of recovery after overfeeding, measurement of circulating hormones.** 17–21-week-old chow-fed male MC4R KO mice on a C57BL/6 J background were overfed as described (Supplementary Data 1). One mouse was excluded before the beginning of the study due to its low body weight (26 g). Mice were divided into experimental groups: control ($n$ = 6) and ExpOF ($n$ = 5) matched for baseline body weight (control = 43.5 ± 3.0 g, ExpOF = 41.3 ± 1.7 g) and baseline daily food intake (control = 19.5 ± 1.2 kcal, ExpOF = 20.4 ± 1.3 kcal).

**Study 8: Experimental overfeeding of FGF21 KO mice—observation of recovery after overfeeding.** 20–30-week-old chow-fed male FGF21 KO mice on a C57BL/6 J background were overfed as described (Supplementary Data 1). FGF21 KO mice were divided into experimental groups: Control ($n$ = 4) and ExpOF ($n$ = 5) matched for baseline body weight (control = 30.6 ± 2.7 g, ExpOF = 29.1 ± 3.3 g), but not for baseline daily food intake (control = 15.4 ± 1.2 kcal, ExpOF = 12.4 ± 0.5 kcal), as mice used as control were slightly older than the used for overfeeding. Energy infusions were calculated using baseline intake of mice in ExpOF group. We employed a slightly modified diet without sucrose supplementation (1.5 kcal/mL). Therefore, the macronutrient composition (35E% from fat, 50E% from carbohydrates, 15E% from protein) was slightly different from the composition of the diet used in the rest of the experiments.

**Administration of recombinant proteins**
Experiments were conducted using diet-induced obese (DIO) male C57BL/6 J (Janvier Labs) kept on a high-fat, high-sugar diet (HFD) (58 kcal% fat, #D12331i, Research Diets) from 8 weeks of age. Mice were fed a HFD diet ad libitum for a minimum of 16 weeks and had an average body weight of >45 g before initiation of the study. Mice were single-housed at least one week before the injections and received once-daily sham injections with isotonic saline three times before the injections of the recombinant proteins. Mice were randomized to treatments based on body weight. DLL1 and LGMN human

recombinant protein were obtained from Sino Biological (#11635-H08H) and Cusabio (#EP012903HU), respectively. These recombinant proteins (or PBS as vehicle) were administered as a single injection at 5 pm with concomitant measurements of body weight and food intake at indicated times. Proteins were administered at 1 mg/kg dose in a volume of 5 μL per gram of body weight. The same approach was followed for lean mice that were injected with 1 mg/kg LGMN protein for open field test, except that lean mice were injected in the morning 30 minutes prior to the open field test.

**Behavioral analysis (open-field test)**
An open field test (OFT) was conducted to quantify locomotor activity and anxiety-related behavior in mice introduced into a novel environment[57]. Mice were acclimatized to the behavioral room and housed in the room in open cages for one week before the OFT was conducted. We used a 50 cm × 50 cm × 50 cm arena, dividing it into a border and a center zone. The OFT was carried out on lean chow-fed C57BL/6 J mice at 14 weeks of age. Animals were injected with LGMN 1 mg/kg or PBS 30 minutes prior to the test. The arenas for treated mice were interchanged, so all arenas were used for both treatments (LGMN or vehicle). Experiments were carried out for 20 min in the light phase from 09:00–13:00, where all 20 minutes in the arena were used for the analysis. We determined the amount of time spent in the center square, compared to the sides, velocity, and distance traveled using Noldus EthoVision XT 17™ software (Noldus, NL). $n$ = 10 mice per group.

**Blood and tissue analysis**
**Gene expression analysis (RNA extraction, cDNA synthesis, and qPCR).** Gene expression profiling in epidydymal white adipose tissue (eWAT), inguinal WAT (iWAT), interscapular brown adipose tissue (BAT), liver, and quatriceps femoris muscle was performed in control and ExpOF mice at d14 and d14 + 3 ($n$ = 6). Tissues were quickly dissected in the morning without fasting the animals, flash-frozen on dry ice, and stored at −80 °C until analysis. Total RNA was isolated from tissues with phenol/chloroform extraction using QIAzol reagent (Qiagen, Germany) and RNeasy Lipid Tissue Mini Kit (Qiagen, Germany) following the instructions provided by the manufacturer. After extraction, RNA concentration and purity were measured using a NanoDrop 2000 (Thermo Fisher Scientific, USA). A total of 500 ng of RNA was converted into cDNA using Superscript III (Thermo Fisher Scientific, USA), following the instructions from the manufacturer. Quantitative PCR (qPCR) was performed using Precision plus qPCR Mastermix containing SYBR green (Primer Design, UK)). For primer sequences, see Supplementary Data S3. Quantification of mRNA expression was performed according to the delta-delta Ct method. All results were normalized to housekeeping genes *Rplp0* (iBAT), *Hprt* (iWAT, muscle, and liver), or *Rpl13a* (eWAT). Primers are stated in Supplementary Data 5.

**Plasma profiling.** Mouse plasma samples were analyzed using ELISA assays to measure insulin (Ultra-Sensitive Mouse Insulin ELISA Kit, #90080, Crystal Chem, USA), leptin (Mouse/Rat Leptin Immunoassay, #MOB00B, R&D Systems, USA), total ghrelin (Rat/Mouse Total Ghrelin, #EZRGRT-91K, EMD Millipore, USA), GDF15 (Rat/mouse GDF15 Quantikine ELISA kit, #MGD150, R&D Systems, USA) and FGF21 (Fibroblast Growth Factor 21 Mouse/Rat ELISA, #RD291108200R, Bio-Vendor R&D, Czech Republic) using manufacturer instructions. Plasma was diluted 1:5 for total ghrelin measurement, 1:20 (1:60 for ExpOF d14 samples) for leptin measurements, 1:10 for FGF21 (1:3 for controls) and 1:5 for GDF15 measurements. Blood glucose was measured from tail blood in awake mice using a glucometer. Plasma total cholesterol (Infinity Cholesterol Liquid Stable Reagent, #TR13421, Thermo Fisher Scientific, USA), total glycerol (triglycerides) (Infinity Triglyceride Liquid Stable Reagent, #TR22421, Thermo Fisher Scientific, USA), and

free fatty acids (HR Series NEFA-HR(2), #434-91795, #436-91995, #270-77000, Fujifilm Wako Chemicals Europe) concentrations were measured by using commercially available kits.

**Olink plasma proteomics.** Protein levels were measured in mouse plasma ($n = 6$) using the Mouse Exploratory panel from the Olink platform (Olink Proteomics, Uppsala, Sweden), covering a total of 92 distinct protein assays. Olink proteomics is based on a proximity extension assay, where oligonucleotide-labeled antibody probe pairs are allowed to bind to their respective targets in the sample in 96-well plate format. Data are presented as log2-transformed units NPX. Higher NPX values correspond to a higher protein expression. The log2 NPX data was transformed into linear data for statistical analysis. Olink proteomics data were analyzed using Metaboanalyst 5.0[58,59]. A comprehensive list of all proteins measured is available in Supplementary Data 2.

**Transcriptomic analysis by RNA sequencing.** Total RNA was extracted from frozen mouse hypothalami ($n = 6$) of four experimental groups ($n = 6$: control d14, ExpOF d14, control d14 + 3, ExpOF d14 + 3) using RNeasy mini kit (Qiagen) according to the manufacturers' protocol. Messenger RNA-sequencing libraries were prepared using the Illumina TruSeq Stranded mRNA protocol (Illumina). Poly-A containing mRNAs were purified by poly-T attached magnetic beads, fragmented, and cDNA was synthesized using SuperScript III Reverse Transcriptase (Thermo Fisher Scientific). To prime for adapter ligation cDNA was adenylated, and after a clean-up using AMPure beads (Beckman coulter), the DNA fragments were amplified using PCR followed by a final clean-up. Libraries were quality-controlled using a Bioanalyzer instrument (Agilent Technologies) and subjected to 51-bp paired-end sequencing on a NovaSeq 6000 (Illumina). A total of 2.74 billion reads were generated.

The STAR aligner12 v. 2.7.3a was used to align RNA-seq read against the mm10 mouse genome assembly and GENCODE vM22 mouse transcripts[60]. The software program featureCounts v. 1.6.4 was used to summarize reads onto genes[61]. Testing for differential expression was performed using DESeq2 v. 1.30.1[62] with a fitted model of the form -group where the group encoded both genotype and treatment. Contrasts were constructed as described in the DESeq2 manual. Gene Ontology5 enrichments were found using the gseGO function from clusterProfiler v. 3.18.1[63]. Genes were ranked by the test statistics as provided by DESeq2. Only terms with between 10 and 300 genes were investigated. One mouse from the control d14 was removed from the analyses due to poor RNA quality. A full list of genes and pathways can be found in Supplementary Data 3 and 4.

**Whole-brain CD31 staining.** Staining was performed according to the iDISCO protocol[64]. Mice were whole body intracardially perfusion fixated using first 1× phosphate-buffered saline (PBS) with added heparin (15,000 UI/L) followed by 10% Neutral Buffered Formalin (NBF) (CellPath, ref. 1000.5000). Brains were isolated and stored in NBF overnight (ON). Brains were washed 3 × 30 min in PBS (with shaking). Brains were dehydrated in methanol/H2O gradient: 20%, 40%, 60%, 80%, and 100% methanol, each step 1 hour (room temperature). They were further washed in 100% methanol for 1 hour and incubated overnight in 66%DCM (Dichloromethane)/33% methanol at room temperature. The next day, the samples were washed twice in 100% methanol for 30 minutes, cooled down to 4 °C in 1 hour, and bleached in chilled fresh 5% H2O2 in methanol (1 volume 35% H2O2 to six times volume methanol) overnight at 4 °C. The brains were subsequently rehydrated in methanol/PBS series: 80%, 60%, 40%, 20%, with 0.2% Triton X-100, 1 hour each at room temperature. They were washed in PBS with 0.2% Triton X-100 (PTx.2) for 2 × 1 hour at room temperature. The brains were then incubated in permeabilization solution at 37 °C for 3 days. Blocking was carried out in blocking solution (PBS with 0.2% Tween-20

and 10 mg heparin, 2 ml Tween-20, (PTwH)/5%DMSO/3% donkey serum) at 37 °C for 2 days. The samples were hereafter incubated with primary antibody CD31 (Goat anti-CD31, R&D systems cat# AF3628, lot # YZU0120101) in PTwH/5%DMSO/3% donkey serum at 37 °C for 7 days. Next, they were washed in PTwH for 1 × 10 minutes, 1 × 20 minutes, 1 × 30 minutes, 1 × 1 hour, 1 × 2 hours and 1 × 2 days. The brains were then incubated with secondary antibody (AF790 donkey a-goat lot #154504) in PTwH/3% donkey serum at 37 °C for 7 days, followed by washes in PTwH: 1 × 10 minutes, 1 × 20 minutes, 1 × 30 minutes, 1 × 1 hour, 1 × 2 hours and 1 × 3 days. Tissue was cleared in methanol/H2O series: 20%, 40%, 60%, 80%, and 100% for 1 hour each at room temperature. Brains were incubated in 100% methanol overnight and next day for 3 hours (with shaking) in 66%DCM (Dichloromethane)/33% methanol at room temperature and in 100% DCM 15 minutes twice (with shaking) to remove traces of methanol. The samples were finally transferred to DiBenzyl Ether and stored in closed glass vials in dark until imaged.

**Light-sheet imaging and 3D image analysis.** Brains were imaged using a Lavision ultramicroscope system II and MV PLAPO 2× C objective. DBE was used as clearing agent during the acquisition of data. Imaris software was used for 3D visualization of data. Hypothalami of the brains were imaged in high resolution using ×3.2 total magnification. For each sample, the ARC, DMH, and VMH were analyzed in 3D. Vessel length and vessel coverage were quantified using customized software in three steps as described below. First, the high-resolution scan was registered to a light-sheet mouse atlas to obtain region-specific information at the voxel level. Second, the vascular network was segmented, and third, the region information and binary masks were combined to extract the final endpoints. For the atlas mapping, a 25 μm light-sheet mouse brain atlas was utilized[65], from which a sub-volume was extracted, corresponding to the scanned volumes. The scan volumes were downsampled to the same voxel size as the atlas, and hereafter, the atlas volume was registered to the downsampled scan volumes using Elastix[66]. The vascular segmentation was carried out by applying the TubeMap analysis pipeline on the raw scan volumes[67]. From this pipeline, a binary segmentation of the vascular network was obtained, as well as a binary skeleton of the network. In the final step, the registered atlas sub-volume was upsampled to scan resolution and combined with the vascular segmentation and the vascular skeleton to calculate the vascular network length for the three brain regions of interest. Quantifications were done relative to the volume of the area for each mouse. For quantified areas, see Supplementary Fig. 3.

**Histology.** Tissues were post-fixated using 4% paraformaldehyde, dehydrated in ethanol and xylene, and embedded in paraffin. 4 μm slices of tissues were deparaffinized, rehydrated, and stained using Mayer's Haematoxylin and Eosin Y (H&E) staining protocol. Histological images were acquired using light microscopy with the Zeiss Axio observer Colibri 7 inverted microscope, using an Axiocam 702 mono camera with objective plan-APOCROMAT ×20/0.8, ∞/0.17 (Na 0.55 WD 25 mm), and Zen v3.0 software. The adipocyte area of eWAT was measured using the watershed function on ImageJ v1.52[68] software quantified from 3 H&E-stained sections, 4 pictures per section (12 images), per mouse. Adipocyte size in control mice at day 14 and day 14 + 3 was very similar, so the data was pooled for statistical analysis.

## Statistics

Statistical analyses were performed in GraphPad Prism version 9.4 (GraphPad, USA). Statistical test information for each experiment can be found in respective figure legends. All data are presented as mean ± SEM, and findings with $P$ values ≤ 0.05 were considered statistically significant. Statistical analyses of multiple groups at once

were performed using one- or two-way ANOVA followed by Bonferroni's post hoc analysis when appropriate. Exceptions to this analysis are indicated and detailed in the figure legends.

## Reporting summary

Further information on research design is available in the Nature Portfolio Reporting Summary linked to this article.

## Data availability

Hypothalamic bulk RNA-seq data generated in this study have been submitted for GEO and are publicly available as of the date of publication under accession number GSE247825. Targeted proteomics raw data are available in the Supplementary information. All other data generated in this study are provided in the Supplemental information/source data file. Source data are provided with this paper.

## Code availability

The source code used to analyze the RNA-seq data is available at Zenodo: https://doi.org/10.5281/zenodo.10497075.

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

## Acknowledgements

We thank Charlotte Sashi Aier Svendsen, the single-cell omics platform (SCOP), and the Rodent Metabolic Phenotyping Platform (RMPP) for experimental, technical, and bioinformatical assistance. We also thank members of the Clemmensen group for scientific discussions. C.C. is supported by research grants from the Lundbeck Foundation (Fellowship R238-2016-2859) and the Novo Nordisk Foundation (grant numbers NNF17OC0026114, NNF22OC0073778). THP acknowledges the Danish Council for Independent Research (Grant number 8045-00091B). The Novo Nordisk Foundation Center for Basic Metabolic Research is an independent Research Center based at the University of Copenhagen, Denmark, and partially funded by an unconditional donation from the Novo Nordisk Foundation (www.cbmr.ku.dk) (Grant number NNF18CC0034900).

## Author contributions

C.L., S.F., P.R.R., and C.C. conceived the study. C.L. and S.F. developed the experimental overfeeding procedure. C.L and P.R.-R. performed animal surgeries and executed the mouse in vivo studies as well as laboratory analyses with support from S.F. and V.K.V.-P. Surgeries were supported by N.K. & V.V. Bioinformatic and transcriptomic analysis were performed by D.M.R. and T.H.P. iDISCO staining and analysis was performed by G.S., J.L.S. and U.R. C.L., P.R.-R., J.L., and C.C. wrote the manuscript. All co-authors provided input to the manuscript.

## Competing interests

C.C. is a co-founder of Ousia Pharma ApS, a biotech company developing therapeutics for the treatment of metabolic disease. The remaining authors declare no competing interests.
