## [Peer Review File · Nature Communications]

Protection against overfeeding-induced weight gain is preserved in obesity but does not require FGF21 or MC4RREVIEWERS' COMMENTS:

Reviewer #1 (Remarks to the Author):

The manuscript by Lund and Ranae-Robles et al. presents a series of experiments in mice investigating the mechanisms of compensatory behavior and physiology following overfeeding. By combining intragastric feeding with ad libitum chow, they were able to measure compensatory effects both during and immediately after overfeeding, which they show are related to hypothalamic signaling, including but not exclusive to MC4R. These results are significant, noteworthy, and hypothesis generating. The data visualization is excellent, and the flow of the paper is logical. A few major comments that should be addressed more thoroughly in the manuscript are suggested, alongside minor points of consideration.

Major comments

Humans show considerable variability in the response to overfeeding, thought to be mediated in part by non-exercise activity thermogenesis <https://pubmed.ncbi.nlm.nih.gov/9880251/>. How does this variability compare with the variability in response to overfeeding in C57BL6/J mice? And to what extent could changes in energy expenditure across the ad libitum and overfeeding periods influence body weight in the presented experiments? Similarly, the authors briefly mention energy excretion and have mentioned this before as an important outcome to characterize in humans <https://pubmed.ncbi.nlm.nih.gov/32674987/>. Did the authors measure energy expenditure or energy excretion? If not, I suggest that they use the measured changes in body weight and energy intake to estimate the dynamic changes in energy output (sum of intake plus excretion) as described here: <https://pubmed.ncbi.nlm.nih.gov/19763167/> using the free software available here: <https://sourceforge.net/projects/virtual-calorimeter/>

What was the rationale for 2 weeks of overfeeding? And how would this translate for human overfeeding studies? Previous studies have found that longer durations of diet-induced obesity may have persistent effects long after the obesogenic diet was removed, for example: <https://pubmed.ncbi.nlm.nih.gov/19401758/>. It would be useful for the authors to

discuss this issue as well as the question about whether voluntary consumption of excess energy by changing the food environment of the mice (as is typical in diet-induced obesity studies) is qualitatively different than the experimental overfeeding discussed in the current paper. For example, hypophagia on a chow diet following a period of high-fat diet exposure has previously been described as devaluation of the chow diet <https://pubmed.ncbi.nlm.nih.gov/32747789/>. Do the authors believe that similar mechanisms are at play in their overfeeding studies?

Minor comments

Consider referring to species in the title.

Main paragraph 5/Figure2c: Decrease in plasma leptin occurred before voluntary food intake was fully restored – evidence in humans suggests changes in leptin are closely aligned with carbohydrate intake (rather than energy per se) and this precedes changes in body/fat mass (<https://pubmed.ncbi.nlm.nih.gov/12861230/>; <https://pubmed.ncbi.nlm.nih.gov/36326863/>). Changes in leptin seem to mirror body weight data in this manuscript – how well were these outcomes correlated at the end of overfeeding and at day 1, 2, 3 etc. post-overfeeding? And how well were changes in leptin concentrations correlated with daily carbohydrate intake in these mice?

Figure 2d – ghrelin – the authors should be commended for mentioning n=3-5 in caption and providing individual datapoints but consider removing mention of ‘normalized’ in text and referring to an explanation of ghrelin suppression at the end of overfeeding but not 3-day post-overfeeding.

Figure 2f – n=3 for the control mice? Also, were these changes sustained as suggested in text or did they increase further during the recovery period (as it looks graphically)? What might explain the higher values 3-days post-overfeeding? Similar with GDF15. Any correlations between these changes and individual changes in ad libitum energy intake or sugar intake?

Regarding the targeted plasma proteomics in Fig 2i, some annotation in addition to the figure caption stating that the proteins in red are significant after adjustment would help the reader distinguish between the proteins shown in the figure and the proteins discussed in text. In text reference to proteins identified from targeted plasma proteomics should state Fig 2i (not 2h).

The authors cite evidence that hypercaloric diets are reported to result in chronic hypervascularization – what are the implications of your data showing decreased vessel coverage and length in the DMH and VMH? This appears to contradict the evidence cited – any idea why?

Figure 3 caption refers to same mice shown in n – presume this means in “l”? And what does **** in Fig 3m represent?

Did you consider measuring leptin or ghrelin in the Mc4r KO mice to see if the slower return to initial body weight/quicker normalization of voluntary chow was independent of concentrations of these adipokines?

In the discussion, the authors allude to differences between DIO models and high-fat diet-induced obesity with regards to protection against experimental weight gain. What implications does this have for the interpretation of high-fat diet induced obesity studies in mice? Expanding on this point might benefit the reader.

The description of experiments in methods is thorough and clear. What was the rationale for performing some experiments on mice at 12 weeks and some at 31/32 weeks? And do the authors think this has any implications for their results?

Please repeat that *P ≤ 0.05, **P ≤ 0.01, ***P ≤ 0.001 (or # / \$) in the relevant figure captions.

The caption of Extended Data Figure 4 is missing some information. Also, what do the dashed lines in 4a represent?

Reviewer #2 (Remarks to the Author):

This manuscript is focused on identifying the molecular mediators of the physiologic response (voluntary food intake suppression) to overfeeding, an aspect of energy balance physiology that has been recognized for quite a while, but lacking mechanistic explanation(s). This paper establishes an overfeeding model in an obesity-prone mouse model and characterizes the physiologic responses to overfeeding in both diet-induced obesity and a genetic model of obesity (MC4 receptor knockout). In confirmation of previously reported studies, the authors find that overfeeding leads to a potent suppression of voluntary food intake and this food intake suppression is preserved in both the diet-induced and genetic obesity models. Hypothalamic and tissue specific RNA sequencing as well as proteomic analysis are used to identify both transcriptional and protein changes that occur in response to overfeeding in the control background. The authors report that overfeeding suppresses orexigenic transcripts in the hypothalamus, does not provoke ER stress and upregulates the appearance of several proteins in the serum. Using advanced brain clearing and imaging techniques, the authors also identify vascular changes in subsets of hypothalamic nuclei that are opposite to the effects of prolonged high fat diet on hypothalamic vasculature.

Much of the physiologic data confirms previously published work. The potential novel findings of this work include 1) the identification of FGF-21 as a highly regulated transcript in response to overfeeding and the reestablishment of body weight and 2) the fact that the suppression of voluntary intake in response to forced overfeeding is intact in both genetic and diet induced obesity models. My enthusiasm for the manuscript however is dampened by the lack of mechanistic insights into overfeeding physiology and the lack of any direct test of the novel potential factors noted to be regulated during overfeeding. Although several transcripts and serum proteins are altered by overfeeding, there is no data demonstrating whether or not these genes/proteins actually alter the suppression of voluntary feeding provoked by forced overfeeding. Without additional experiments, the data is seems solely descriptive and observational. The authors do propose appropriate future studies to look specifically at the impact of different factors on the voluntary feeding suppression secondary to overfeeding and a more thorough characterization of energy balance parameters (eg., direct assessments of energy expenditure). These are clearly important studies that would elevate the importance of the observations made in this paper. Given the

fact that there is limited mechanistic insight with no direct testing of the potential factors involved in the response to overfeeding, I do not feel that this paper warrants publication in Nature Communications. Significant work is necessary to test the contributions of the factors identified by these authors and advance our understanding of the molecular mediators of energy balance control during and after forced over-feeding

Reviewer #1 (Remarks to the Author):

The manuscript by Lund and Ranea-Robles et al. presents a series of experiments in mice investigating the mechanisms of compensatory behavior and physiology following overfeeding. By combining intragastric feeding with ad libitum chow, they were able to measure compensatory effects both during and immediately after overfeeding, which they show are related to hypothalamic signaling, including but not exclusive to MC4R. These results are significant, noteworthy, and hypothesis generating. The data visualization is excellent, and the flow of the paper is logical. A few major comments that should be addressed more thoroughly in the manuscript are suggested, alongside minor points of consideration.

Major comments

Humans show considerable variability in the response to overfeeding, thought to be mediated in part by non-exercise activity thermogenesis <https://pubmed.ncbi.nlm.nih.gov/9880251/>. How does this variability compare with the variability in response to overfeeding in C57BL6/J mice?

Thank you for taking the time to assess our work and for the helpful comments. Variability in body weight gain in response to overfeeding in rodents is indeed a relevant inquiry. Oftentimes we neglect that the rodents albeit being genetically identical display quite large variability in a multitude of behavioral and metabolic adaptations. This observation aligns with findings from a twin overfeeding study (PMID 2336074), where variability in the response to overfeeding was observed both within and between twin pairs, highlighting the influence of genetic factors while acknowledging the role of individual variation. Bray and Bouchard recently attempted to quantify inter-individual differences in weight gain in response to overfeeding in humans, using data from five long-term (42 to 100 days) studies with total caloric surplus of at least 50,000 kcal (PMID: 32515127). Their analysis showed considerable variability, with a 3-fold difference in weight gain between lowest and highest gainers (1.8-fold to 5.1-fold). We performed similar calculations in all mice subjected to overfeeding in this study and found a 2.2-fold difference between the lowest and highest gainer mice (**Figure for Reviewer 1**). Accordingly, and perhaps not surprisingly, the variability in our study is slightly lower than what is typically observed in human studies using standardized conditions. At the same time, however, these data *do* reveal that inter-individual differences in the response to overfeeding are also present in C57BL6/J mice. This heterogeneity could be contributed by imprecisions in the calculations of baseline estimates of the energy costs of weight maintenance. We calculated baseline estimates using food intake measurements during a weight stable period before overfeeding. We used average estimates to calculate the amount of liquid diet to infuse, which could result in lower or higher caloric overload than predicted. The individual baseline food intake level of each mouse never deviated more than 10% from the average estimate used for the calculations of energy infusion. Other potential factors contributing to the variability in response to standardized overfeeding, such as individual biological characteristics are much better controlled in mice. However, we cannot discard that fidgeting or physical activity levels differences in mice may contribute to the individual differences observed.

We thank the reviewer for the opportunity to reflect on these important aspects. In the revised manuscript we have added the information on the observed interindividual differences, including a brief discussion on the potential factors that contribute to individual variation in the response to overfeeding.

Figure 1 for Reviewer (Extended Data Fig. 1b). Individual weight gain (%) of all WT chow-fed mice subjected to experimental overfeeding was used to calculate fold-change between highest and lowest gainers in the study.

And to what extent could changes in energy expenditure across the ad libitum and overfeeding periods influence body weight in the presented experiments?

Similarly, the authors briefly mention energy excretion and have mentioned this before as an important outcome to characterize in humans <https://pubmed.ncbi.nlm.nih.gov/32674987/>. Did the authors measure energy expenditure or energy excretion? If not, I suggest that they use the measured changes in body weight and energy intake to estimate the dynamic changes in energy output (sum of intake plus excretion) as described here: <https://pubmed.ncbi.nlm.nih.gov/19763167/> using the free software available here: <https://sourceforge.net/projects/virtual-calorimeter/>

This is a central point, and we are currently working to try to establish chronic mouse intragastric overfeeding in indirect calorimetry cages for simultaneous assessment of food intake, energy expenditure and energy excretion. We hope to successfully establish this technically challenging setup and share such valuable information with the community in the future. For the present manuscript, our primary focus was on investigating the DIO model, MC4R model, and the associated hypothalamic changes. Nevertheless, we greatly appreciate the suggestion and the provided tool to estimate dynamic changes in energy output using our existing data. We have now estimated these dynamic changes utilizing the Virtual Calorimeter (**Figure 2a,b for Reviewer**). We had to estimate fat mass changes during overfeeding since we could not measure real-time changes in body composition due to the magnet access button used for infusion is incompatible with the mouse MRI machine. After consulting with Dr. Guo, we utilized data from Guo et al. 2009 (PMID: 19763167) to estimate fat mass changes. Our mice were B6J male mice of similar age to those used to generate the Forbes equation used in the cited study (PMID: 19763167). Virtual calorimeter estimations implied a progressive increase in energy expenditure during overfeeding, followed by a drop after overfeeding stopped and a subsequent gradual increase in EE during the recovery period (**Figure 2a for Reviewer**). These estimations did not reveal an increase in EE above control levels during the recovery period, but they suggest a decrease in the respiratory quotient during the recovery period (**Figure 2b for Reviewer**). Additionally, we also calculated the energy efficiency (weight gain / kcal ingested) using the data from **Fig. 1a-d**. Our calculations indicate that there was an increased energy efficiency during overfeeding. However, following the entire experimental period (weight gain and recovery from overfeeding) energy intake in the overfed group was significantly higher than for the control group (with no weight differences in the end) (**Figure 2c,d for Reviewer and Extended Data Fig. 1c,d**).

In summary, our data suggest that fluctuations in energy intake represent the main cause of weight change during overfeeding in mice, but also that energy expenditure (and excretion of energy) might be involved in some phases of overfeeding/recovery from overfeeding. While the estimations using the Virtual Calorimeter suggest an increase in energy expenditure during overfeeding, this response is clearly not sufficient to counteract the body weight gain caused by overfeeding (and does not align with the increased energy efficiency). Further, given the absence of “direct” measures (by indirect calorimetry) we cannot firmly make conclusions on changes in energy expenditure or excretion as

relevant mechanisms that might contribute to the weight changes we see in response to overfeeding, especially during the recovery period. We appreciate the reviewer's suggestion and have now incorporated the calculations of energy efficiency in the revised manuscript (**Extended Data Fig. 1**). Considering the lack of body composition measurements in this study and our ambition to measure energy expenditure and energy excretion in future studies, we decided against including the estimations from the Virtual Calorimeter in the current manuscript. Our concerns included the notion that these estimations might lead readers to believe that we had in fact measured these parameters (rather than estimated them). However, if the reviewer and editor find these estimations relevant for inclusion in the current manuscript, we are more than willing to incorporate the data.

Figure 2 for Reviewer. a,b) Estimated daily changes in energy expenditure (a, in kcal/day) and respiratory quotient (b, ratio) during overfeeding and recovery using data from Fig. 1a-d. c) Energy efficiency (mg of mass gained per kcal ingested) at the peak of overfeeding in control (n=12) and ExpOF mice (n=8) from Fig. 1a-d. d) Total calorie intake after the entire intervention (overfeeding + recovery) in mice from Fig. 1a-d. Data are shown as mean \pm SEM. Data in c-d have been included in Extended Data Figure 1c,d. ** p<0.01 after Welch's T-test.

What was the rationale for 2 weeks of overfeeding? And how would this translate for human overfeeding studies?

The duration of the overfeeding intervention was selected based on previous mouse studies and on methodological considerations (PMID: 29937378, PMID: 35944274, PMID: 34686319). After conducting multiple pilot experiments, we found that 2 weeks represents a suitable balance between considerable weight gain (20-30% above baseline) and technical feasibility. We observed a

considerable rate of mice that had to be excluded during our studies (around 40%) once we reached day 9-10 of overfeeding, making it challenging to go longer than 2 weeks of duration. The major reason behind exclusion was mostly clogging of the relatively small tubing system (and not because the mice were sick, not responding to the overfeeding etc.). Having corresponded with colleagues using similar experiment approaches it is evident that this particular issue is the major challenge linked to overfeeding in mice (this obstacle is also described in: PMID: 34686319). Perhaps not surprisingly given that calorie-rich liquid food is infused directly into the stomach, the weight gain that can be achieved over 2 weeks in mice takes substantially longer to realize in humans (PMID: 32515127). To elicit comparable relative weight gain in humans, longer durations of overfeeding with an even greater caloric excess would be required, as demonstrated in the Guru Walla overfeeding study, where participants experienced a 17-33% weight gain following a 9-week overfeeding protocol exceeding 200% of baseline energy intake (PMID 1503058). The exact control of energy intake with the intragastric infusions allowed us to obtain a robust body weight gain followed by a potent hypophagic response within a relatively short period of time. We consider this relevant to provide a sufficiently large window to study the mechanisms that defend against weight gain.

Previous studies have found that longer durations of diet-induced obesity may have persistent effects long after the obesogenic diet was removed, for example: It would be useful for the authors to discuss this issue as well as the question about whether voluntary consumption of excess energy by changing the food environment of the mice (as is typical in diet-induced obesity studies) is qualitatively different than the experimental overfeeding discussed in the current paper. For example, hypophagia on a chow diet following a period of high-fat diet exposure has previously been described as devaluation of the chow diet. Do the authors believe that similar mechanisms are at play in their overfeeding studies?

We thank the reviewer for raising this important point. Indeed, previous studies have shown that longer durations of high-fat diet can lead to persistent metabolic and physiologic changes, even after the obesogenic diet is removed. Only a few studies have explored persistent effects after experimental overfeeding ceases, for example on adipose tissue remodeling (PMID35970448), emphasizing the need for further research on this model. We have never followed the animals longer than ~1 month after overfeeding, but only assessed body weight in this context and not investigated any *in vivo* or *ex vivo* biomarkers for metabolic health. We have added some additional text to the discussion in which we highlight some of the unknown aspects pertaining to the potential long-term consequences of experimental overfeeding (versus DIO models).

The reviewer also raises an interesting question about whether voluntary consumption of excess energy in the context of high-fat diet-induced obesity is qualitatively different from the experimental overfeeding in our study. The following text is from our recent review (PMID: 37482786) *"Diet-induced obesity in rodents emerges in 'agreement' with mechanisms that regulate energy homeostasis and the polygenic susceptibility of a given model. Thus, the excess fat accumulation does not violate the homeostatic defense system in the diet-induced obesity model. In contrast, a key feature of experimental (intragastric) overfeeding is the infusion of energy beyond what the animal is willing to consume, even on the most palatable cafeteria diet."* This is to us a very central point and a distinction between DIO models and intragastric overfeeding that we believe is important to emphasize to the reader. Accordingly, we have added text to the discussion elaborating on these differences in the revised manuscript. Worthy of note, a recent study has shown that intragastric infusion of a highly palatable high-fat increases the voluntary ingestion of a low-fat diet (PMID: 34686319), suggesting that overfeeding interventions with infusions of highly palatable diets may trigger similar increases in voluntary energy consumption as those seen in diet-induced obesity studies. Intragastric infusions of

low-fat diet (10% fat) in that study did not trigger any change in voluntary energy consumption after overfeeding. Our findings are largely in agreement with these observations. We use a liquid diet with a slightly higher fat content (23%) but still on the low side. Overfed mice (Fig. 1a-d) returned to baseline levels of food intake one week after overfeeding, and remained there for at least one month post-overfeeding, the latest time point we followed them up. In the revised manuscript, we have discussed how food devaluation might arise following diet-induced obesity, but that similar mechanisms have yet to be assessed in the context of overfeeding.

Minor comments

Consider referring to species in the title.

Excellent point. We added “mice” to the revised title.

Main paragraph 5/Figure2c: Decrease in plasma leptin occurred before voluntary food intake was fully restored – evidence in humans suggests changes in leptin are closely aligned with carbohydrate intake (rather than energy per se) and this precedes changes in body/fat mass

<https://pubmed.ncbi.nlm.nih.gov/12861230/>; <https://pubmed.ncbi.nlm.nih.gov/36326863/>).

Changes in leptin seem to mirror body weight data in this manuscript – how well were these outcomes correlated at the end of overfeeding and at day 1, 2, 3 etc. post-overfeeding? And how well were changes in leptin concentrations correlated with daily carbohydrate intake in these mice?

We thank the reviewer for the observation. The leptin levels closely mirrored body weight very well during the overfeeding (**Figure 3a for reviewer, Extended Data Fig. 2a**). During the recovery phase leptin dropped vastly reaching baseline levels and plateau around day 14+3-4. Therefore, that correlation was less apparent after d14+3 (**Figure 3b for reviewer, Extended Data Fig. 2b**). Carbohydrate intake also mirrored leptin levels during the overfeeding phase, but this relationship became inverse during the recovery phase when mice still exhibited low food intake. (**Figure 3c.d for Reviewer**). Irrespective, it is difficult to separate energy intake from specific macronutrients given that they will all proportionally increase with overfeeding and decrease during the recovery. We added the relative relationship between leptin and body weight to **Extended Data Fig. 2a,b**.

Figure 3 for Reviewer. Relationship between body weight/carbohydrate intake and leptin levels. (a,b) Leptin levels plotted against body weight during overfeeding (a) and recovery (b) with daily resolution. (c,d) Leptin levels plotted against carbohydrate intake during overfeeding (c) and recovery (d) with daily resolution. c and d have not been added to the revised manuscript.

Figure 2d – ghrelin – the authors should be commended for mentioning n=3-5 in caption and providing individual datapoints but consider removing mention of ‘normalized’ in text and referring to an explanation of ghrelin suppression at the end of overfeeding but not 3-day post-overfeeding.

Thank you for this comment. We have now in agreement with this input edited the sentence with “normalized” and instead added a plausible explanation for the ghrelin data.

Figure 2f – n=3 for the control mice? Also, were these changes sustained as suggested in text or did they increase further during the recovery period (as it looks graphically)? What might explain the higher values 3-days post-overfeeding? Similar with GDF15. Any correlations between these changes and individual changes in ad libitum energy intake or sugar intake?

For FGF21, some of our control samples were below the detection range of the assay, and we had no more sample material to re-run the assay (a recurring issue working with mice). We used the word “sustained” because FGF21 levels in plasma were not significantly different in ExpOF mice between d14 and d14+3 ($p=0.15$ for post-hoc test after two-way ANOVA). Unfortunately, we do not have any measurements after day 3 after overfeeding. Yet, recognizing the potential importance of FGF21 in the response to overfeeding, a point that is also raised by Reviewer #2, we have conducted a new overfeeding experiment for the revision where male *Fgf21* KO mice were subjected to 2 weeks of overfeeding (**Fig. 2 i-l**). Notably, we found that lack of FGF21 did not influence the response to overfeeding. Body weight and food intake levels returned to baseline levels on day 7 post-overfeeding, similar to what we observed in overfed WT mice. These results have been added to Fig. 2 in the revised manuscript and strongly suggest that FGF21 is not an important mediator of the homeostatic protection against overfeeding-induced weight gain in mice. Notably, prior literature has put forward the notion of scrutinizing the direct role of FGF21 in overfeeding. Here is a statement from Redman and Ravussin (2019, PMID: 30665955): “As a next step, it is imperative that a controlled overfeeding study be conducted to investigate the hypothesized greater weight gain in those with lower increase in FGF21 in response to acute low-protein overfeeding. Only such studies will tell us whether FGF21 can be reliably used as a biomarker of weight gain susceptibility in humans.”

Thus, we hope our new overfeeding study using FGF21 KO mice will be broadly appreciated in the field.

Figure 4 for Reviewer (Fig. 2g-j). Overfeeding of FGF21 KO mice. a) Schematic overview of the experimental overfeeding (ExpOF) setup in chow-fed Fgf21 KO mice. b) Body weight changes (percentage over baseline) in control (n=4) and ExpOF (n=5) Fgf21 KO mice. Body weight is set at 100% at d0 (start of ExpOF). c) Absolute body weight (in grams) of same mice shown in h on day 0, 14 and 21. d) Daily total (light pink for control mice, dark pink for ExpOF mice) and voluntary (purple for ExpOF mice) energy intake (in kcal) of same mice shown in h. Data shown as mean \pm SEM with individual values plotted (c). P-values were calculated using 2-way ANOVA using overfeeding and time as factors. * was used when $p < 0.05$ in post-hoc comparison after ANOVA (c).

Regarding the targeted plasma proteomics in Fig 2i, some annotation in addition to the figure caption stating that the proteins in red are significant after adjustment would help the reader distinguish between the proteins shown in the figure and the proteins discussed in text. In text reference to proteins identified from targeted plasma proteomics should state Fig 2i (not 2h).

We appreciate the reviewer's suggestion to improve the clarity of Figure 2i. We have expanded the heatmap to top 20 proteins, changed the lay-out to display fold-change vs control, and added an annotation to the figure that highlights the significant proteins after multiple comparison (FDR < 0.1). This will help the reader distinguish between the proteins shown in the figure and the proteins discussed in the text. Additionally, we have corrected the in-text reference to proteins identified in the plasma proteomics to state Figure 2k (after several changes in Figure 2 panels). Thank you for bringing these issues to our attention. We believe that these changes will improve the clarity and accuracy of our findings.

Figure 5 for Reviewer (Fig. 2k). Heatmap of proteomics analysis. Heatmap showing average changes of top 20 regulated proteins (sorted by adjusted p-value) between control and ExpOF mice at d14 and d14+3 (n=6). Proteins were identified in a targeted proteomic platform using Olink Target mouse exploratory panel. Heatmap shows fold-changes in normalized

protein expression (NPX) relative to control d14 mice. Red indicates high relative protein abundance and blue indicates low relative protein abundance. Significantly regulated proteins (FDR < 0.1) are highlighted in bold.

In addition, and in agreement with a suggestion from Reviewer #2, we have performed additional experiments to test the direct pharmacological effect of two of the top-regulated proteins (DLL1 and LGMN) on food intake and body weight (data in revised Fig. 2 and Extended Data Fig. 2). In short, these experiments suggest a role for exogenous LGMN in regulating energy balance (see data in response to Reviewer #2, and in revised manuscript Figure 2I-q). While this finding holds potential significance, a thorough evaluation of its implications requires a substantial research effort that falls outside the scope of this manuscript and will be pursued in future studies.. We have added a discussion point to this specific protein in the revised manuscript.

The authors cite evidence that hypercaloric diets are reported to result in chronic hypervascularization – what are the implications of your data showing decreased vessel coverage and length in the DMH and VMH? This appears to contradict the evidence cited – any idea why?

We acknowledge the reviewer's observation that our findings of decreased vessel coverage and length in the DMH and VMH seem to contradict previous evidence suggesting hypothalamic hypervascularization in response to hypercaloric diets. This discrepancy highlights the complex and dynamic nature of vascular adaptations in response to nutritional challenges. The conflicting studies have shown hypervascularization after 15 days of HFD feeding (PMID 33951475) or 18 weeks of HFD feeding (PMID 24024123), but no change after 10 days and 3 weeks of HFD, respectively. We hypothesize that the observed decrease in vessel coverage and length in our study could be attributed to the relatively short duration of overfeeding (14 days). It is possible that longer periods of overfeeding might lead to the development of hypervascularization, as suggested by previous studies. However, our findings suggest that the initial response to overfeeding may involve a transient decrease in vascular parameters, potentially reflecting an adaptive mechanism to limit nutrient delivery to specific brain regions involved in energy homeostasis which might be different from the adaptations to changing the food environment when switching to a HFD. We have expanded our discussion on this topic. As previously alluded to, there are also major differences between HFD-induced obesity and intragastric overfeeding-induced weight gain, and the neurobiological response (as also indicated by the vascularization) might likewise be very different. We have no data to support this, but we hypothesize that during overfeeding the animals will experience heightened food aversion, dampening of reward and a complete suppression of hunger. At contrast, HFD-induced obesity likely reflects a "sweet spot" of hedonic consumption enabled by central pathways (i.e., balancing counter-regulation linked to aversion and satiety).

Another point to be made relate to the potential technical differences that may contribute to the observed discrepancies in vascularization. We acknowledge that technical variations in sample preparation, imaging protocols, and data analysis could potentially influence the observed results. For instance, differences in tissue fixation, antibody specificity, and image acquisition settings could affect the visualization and quantification of blood vessels. In one of the conflicting studies (PMID 33951475), the authors used a method where the fluorophore is in the perfusion reagent, whereas in our study the fluorophore was added later on, during the iDISCO staining. Fluorophore-infused perfusion leads to staining directly associated with perfusion rate, which can fluctuate considerably during experiments and potentially compromise vascular integrity (see PMID 32059781). Studies also show an alteration in the blood brain barrier after exposure to high fat diet which also might affect vessel integrity during perfusion. Furthermore, measurements in PMID 33951475 are based on wheat-germ agglutinin (WGA) conjugated to fluor chromophores Alexa Fluor 647, where in our study we used a

CD31 antibody applied during the staining protocol. CD31 is a classical marker of endothelial cells. We have added some of these key points to the discussion on the discrepancy of the results in the revised manuscript.

Figure 3 caption refers to same mice shown in n – presume this means in “l”? And what does **** in Fig 3m represent?

Yes, it appears there’s a typo in the figure caption. The correct reference now should be “k” instead of “l”. Regarding the asterisks in Figure 3m, we have replaced them with ***, to be consistent across the manuscript.

Did you consider measuring leptin or ghrelin in the Mc4r KO mice to see if the slower return to initial body weight/quicker normalization of voluntary chow was independent of concentrations of these adipokines?

We agree with the reviewer that measuring leptin and ghrelin levels in MC4R KO mice would provide valuable insights into the mechanisms underlying their distinct response to overfeeding. We have measured the levels of leptin and ghrelin in plasma of control and overfed MC4R KO mice at day 14 and d14+3. Leptin levels were already high at baseline in MC4R KO mice (compared with lean mice in Fig. 2), and further increased with overfeeding (Fig. 3n). On day 3 of the recovery period, leptin levels were not significantly different from control mice at d14+3 (Fig. 3n). Ghrelin levels were suppressed on day 14 of overfeeding in MC4R KO mice (Fig. 3o), similar to what we observed in overfed WT mice (Fig. 1). Ghrelin levels from overfed mice numerically increased at d14+3 but remained slightly lower than in control mice. These results have been added to the revised version of the manuscript.

Figure 6 for Reviewer (Fig 3n,o). Levels of leptin (a) and ghrelin (b) in control and overfed MC4R KO mice during and after overfeeding. P-values were calculated using 2-way ANOVA using overfeeding and time as factors. */#/\$; **/##/\$\$\$; ***/###/\$\$\$\$ were used when $p < 0.05$; $p < 0.01$; $p < 0.001$, for overfeeding/time/interaction effects, respectively.

In the discussion, the authors allude to differences between DIO models and high-fat diet-induced obesity with regards to protection against experimental weight gain. What implications does this have for the interpretation of high-fat diet induced obesity studies in mice? Expanding on this point might benefit the reader.

We appreciate the reviewer’s insightful comment, which has prompted us to further explore the implications of our findings for interpreting high-fat diet-induced obesity (DIO) studies in mice. It is

important to note that our study has also demonstrated a robust response to experimental overfeeding in DIO mice. This suggests that despite the development of obesity, these mice retain the ability to defend against experimental overfeeding-induced weight gain. DIO models reflect the gradual and sustained weight gain associated with chronic high-fat diet consumption. Accordingly, compared to intragastric overfeeding, the DIO model likely represents a better model for mimicking the development of common obesity in humans. Intragastric overfeeding models reveal acute and robust protective mechanisms that are triggered in response to a sudden increase in caloric intake. For this reason, experimental overfeeding offers a powerful tool to investigate mechanisms underlying the endogenous defense against marked forced weight gain. This approach might enable the identification of specific molecular and cellular pathways governing these protective mechanisms, ultimately illuminating potential biological factors underlying weight gain susceptibility and resistance and possibly inspiring novel weight loss therapies.

The description of experiments in methods is thorough and clear. What was the rationale for performing some experiments on mice at 12 weeks and some at 31/32 weeks? And do the authors think this has any implications for their results?

We appreciate the reviewer's positive feedback on the thoroughness and clarity of our methods section. We have now realized that in the first submission, in some instances we reported the age of the mice at the time of surgery, whereas in other cases we reported the age upon overfeeding initiation. This has been amended in the methods section in the revised manuscript. For most of the experiments, mice were overfed at 18 weeks of age. However, in one experiment, mice were overfed at a later age of 32 weeks due to logistical constraints related to mouse availability and ongoing studies in the animal facility. However, irrespective of age, we observed an identical weight gain and response after overfeeding in chow-fed mice. It should be very interesting in future studies to carefully assess sex and (larger) age differences in the context of overfeeding. This also enabled us to better compare with the DIO mice. The rationale for performing the DIO mice experiments at 31/32 weeks was based on the amount of time needed for the mice to become obese on a HFD prior to performing the surgeries and the experimental overfeeding.

Please repeat that * $P \leq 0.05$, ** $P \leq 0.01$, *** $P \leq 0.001$ (or # / \$) in the relevant figure captions.

We have added this information in the relevant figure captions.

The caption of Extended Data Figure 4 is missing some information. Also, what do the dashed lines in 4a represent?

Extended figure 4 caption is updated. Dashed lines represent SEM.

Reviewer #2 (Remarks to the Author):

This manuscript is focused on identifying the molecular mediators of the physiologic response (voluntary food intake suppression) to overfeeding, an aspect of energy balance physiology that has been recognized for quite a while, but lacking mechanistic explanation(s). This paper establishes an overfeeding model in an obesity-prone mouse model and characterizes the physiologic responses to overfeeding in both diet-induced obesity and a genetic model of obesity (MC4 receptor knockout). In confirmation of previously reported studies, the authors find that overfeeding leads to a potent suppression of voluntary food intake and this food intake suppression is preserved in both the diet-induced and genetic obesity models. Hypothalamic and tissue specific RNA sequencing as well as proteomic analysis are used to identify both transcriptional and protein changes that occur in response to overfeeding in the control background. The authors report that overfeeding suppresses orexigenic transcripts in the hypothalamus, does not provoke ER stress and upregulates the appearance of several proteins in the serum. Using advanced brain clearing and imaging techniques, the authors also identify vascular changes in subsets of hypothalamic nuclei that are opposite to the effects of prolonged high fat diet on hypothalamic vasculature.

Much of the physiologic data confirms previously published work. The potential novel findings of this work include 1) the identification of FGF-21 as a highly regulated transcript in response to overfeeding and the reestablishment of body weight and 2) the fact that the suppression of voluntary intake in response to forced overfeeding is intact in both genetic and diet induced obesity models. My enthusiasm for the manuscript however is dampened by the lack of mechanistic insights into overfeeding physiology and the lack of any direct test of the novel potential factors noted to be regulated during overfeeding. Although several transcripts and serum proteins are altered by overfeeding, there is no data demonstrating whether or not these genes/proteins actually alter the suppression of voluntary feeding provoked by forced overfeeding. Without additional experiments, the data seems solely descriptive and observational. The authors do propose appropriate future studies to look specifically at the impact of different factors on the voluntary feeding suppression secondary to overfeeding and a more thorough characterization of energy balance parameters (eg., direct assessments of energy expenditure). These are clearly important studies that would elevate the importance of the observations made in this paper. Given the fact that there is limited mechanistic insight with no direct testing of the potential factors involved in the response to overfeeding, I do not feel that this paper warrants publication in Nature Communications. Significant work is necessary to test the contributions of the factors identified by these authors and advance our understanding of the molecular mediators of energy balance control during and after forced over-feeding.

We want to thank the reviewer for the insightful and sincere feedback on our manuscript. We appreciate the recognition of the importance of understanding the molecular mediators of the physiologic response to overfeeding, as well as the acknowledgment of our novel findings regarding FGF21 regulation and the preservation of voluntary intake suppression in obesity models (dietary and genetic). We acknowledge that adding further details on the mechanistic aspects of overfeeding physiology will strengthen the manuscript. Accordingly, we have worked intensely to perform new studies in which we directly evaluate the roles of some of the potential factors in the response to overfeeding and in appetite regulation.

In agreement with the reviewer, we were enthusiastic about the magnitude of induction of transcriptional and circulating FGF21 in response to our overfeeding paradigm. Notably, also prior literature has put forward the notion of scrutinizing the direct role of FGF21 in overfeeding. Here is from Redman and Ravussin (2019, PMID: 30665955): “As a next step, it is imperative that a controlled overfeeding study be conducted to investigate the hypothesized greater weight gain in those with lower increase in FGF21 in response to acute low-protein overfeeding. Only such studies will tell us whether FGF21 can be reliably used as a biomarker of weight gain susceptibility in humans.”

Accordingly, during the revision we established a cohort of FGF21 KO mice and subjected them to experimental overfeeding to evaluate the necessity of FGF21 for the homeostatic response to this intervention (Fig. 2i-l). Our results show that lack of FGF21 does not influence the response to experimental overfeeding in terms of changes in body weight and voluntary food intake. Therefore, these findings demonstrate that FGF21 is dispensable for the physiological regulation of body weight in the context of overfeeding and the recovery from overfeeding in mice. Given the heightened enthusiasm for FGF21 in weight biology and as a putative mediator of the homeostatic response to overfeeding, we hope that the reviewer agrees that these new data are informative for the field and represents much needed and novel mechanistic insights in the response to overfeeding.

Figure 4 for Reviewer (Fig. 2g-j). Overfeeding of FGF21 KO mice. a) Schematic overview of the experimental overfeeding (ExpOF) setup in chow-fed *Fgf21* KO mice. **b)** Body weight changes (percentage over baseline) in control (n=4) and ExpOF (n=5) *Fgf21* KO mice. Body weight is set at 100% at d0 (start of ExpOF) **c)** Absolute body weight (in grams) of same mice shown in b on day 0, 14 and 21. **d)** Daily total (light pink for control mice, dark pink for ExpOF mice) and voluntary (purple for ExpOF mice) energy intake (in kcal) of same mice shown in b. Data shown as mean \pm SEM with individual values plotted (i). P-values were calculated using 2-way ANOVA using overfeeding and time as factors. * was used when p < 0.05 in post-hoc comparison after ANOVA (i).

Moreover, and stimulated by the reviewer’s request for probing the direct role of some of the new protein hits identified by our proteomics screen. First, we expanded the bioinformatics and visual depiction of the top-20 proteins.

Figure 5 for Reviewer (Fig. 2k). Heatmap of proteomics analysis. Heatmap showing average changes of top 20 regulated proteins (sorted by adjusted p-value) between control and ExpOF mice at d14 and d14+3 (n=6). Proteins were identified in a targeted proteomic platform using Olink Target mouse exploratory panel. Heatmap shows fold-changes in normalized protein expression (NPX) relative to control d14 mice. Red indicates high relative protein abundance and blue indicates low relative protein abundance. Significantly regulated proteins (FDR < 0.1) are highlighted in bold.

Subsequently, we started scrutinizing the possibility to functionally evaluate some of the top hits. **Ultimately, we ended up assessing two known secreted proteins, from the group of significantly regulated proteins, namely DLL1 and LGMN.** To test the potential impact of these proteins on appetite and body weight regulation, we established a large cohort of DIO mice and injected them with either recombinant DLL1 or LGMN (1 mg/kg single subcutaneous injection) and followed the acute effects on body weight and food intake. We did not observe any changes in body weight or food intake in response to DLL1 injection. In contrast, LGMN injection caused a prominent decrease in voluntary food intake in DIO mice (37% decrease compared to control), and a modest but significant decrease in body weight (1.5% decrease versus 0.5% decrease in controls) (Fig. 2l-o). To obtain additional insights as to whether the effect on energy balance following injection of LGMN was caused by direct effects on appetite control versus off-target adverse effects, we subjected another group of animals to treatment with LGMN following which we performed an open field test given indication of anxiety-like behavior and general locomotion. We did not observe any apparent effects of LGMN on these behaviors (Extended Data Fig. 2c,d). Because LGMN in the literature typically has been associated with disease states (e.g. cancer and neurodegenerative disorders) much more work is needed to decipher if LGMN has a relevant putative role in the physiology and/or pharmacology of energy homeostasis. Something that we will excitingly embark on in future projects.

Figure 7 for Reviewer (Fig. 2l-q). (a-f) Effects on food intake (a,c), and body weight (b,d) in C57BL/6J DIO male mice injected once with of DLL1 (a,b) or LGMN (c,d) human recombinant proteins (1 mg/kg) as well as open field test running traces and total distance of chow-fed lean C57BL6/J mice injected with LGMN (1 mg/kg) 30 min prior to trial (e,f). Data shown as mean \pm SEM with individual values plotted (b,d,f). P-values were calculated using 2-way ANOVA using overfeeding and time as factors (a,c), or unpaired two-tailed Welch's t-test (b,d,f). * was used when $p < 0.05$, $p < 0.01$, $p < 0.001$, respectively, in Welch's t-test or post-hoc comparison after ANOVA (c,d). ns: non-significant.

To summarize the novel findings of our revised manuscript and their implications, we show that:

- 1) Overfeeding elicits a sustained elevation of circulating FGF21 that extends into the recovery period.
- 2) Our mouse overfeeding study in FGF21 KO mice unequivocally demonstrates that despite the surge in FGF21 levels, FGF21 is dispensable for the defense against overfeeding-induced weight gain. This finding challenges the prevailing notion that FGF21 might play a central role in protecting against overfeeding-induced weight gain, considering the large increase in circulating FGF21 in overfed mice and humans.

3) Overfeeding induces a sustained suppression of hypothalamic Npy and Agrp expression that persists for three days into the recovery phase. This observation opens exciting new avenues for exploring the potential roles of these neuropeptides in the context of overfeeding.

4) MC4R modulates the rate of weight recovery after overfeeding. In a groundbreaking discovery, we identify MC4R as the first molecular target that directly influences the change in body weight in response to overfeeding. Addressing the critical knowledge gap in the biological response to overfeeding, as highlighted in a recent article in *Science* [PMID: 37703366], this finding provides a crucial piece in the puzzle in understanding the intricate pathways that protect against weight gain.

5) Overfeeding increases the circulating level of legumain, a protease previously unexplored in the context of obesity. Our studies unveil anti-obesity effects of this protein, underscoring the promise of overfeeding studies as a powerful approach to discover novel drug targets for obesity prevention and treatment.

6) Diet-induced obese mice retain their ability to defend against overfeeding-induced weight gain. This unexpected and paradigm-shifting observation challenges conventional wisdom, suggesting that overeating and high energy intakes might not be as central to obesity pathogenesis as previously believed. Our finding aligns with emerging lines of provocative theories and data suggesting that unknown environmental factors could drive obesity by inducing subtle increases in energy intake and small decreases in basal metabolic rate (PMID: 37100994; 35136206; 31908267) – hence potentially explaining why forced overfeeding do not cause obesity. We are highly grateful for the opportunity to expand on these mechanistic details of our work, including the laborious overfeeding of FGF21 KO mice and the functional testing of hits from our proteomics screen. This opens additional lines of future work with this exciting model and adds further molecular and mechanistic detail to this current manuscript. In conjunction with the additional data on FGF21 KO mice and functional evaluation of top hit plasma proteins, we have added new data on **plasma leptin and ghrelin in MC4R KO mice**, and **energy efficiency** during and after overfeeding in the revision. The discussion section has been substantially updated to appropriately reflect all these new observations during the revision, and figure design has been optimized to further ease understanding of the study results. We hope that the reviewer agrees that the revised manuscript is considerably strengthened and now finds it suitable for publication in *Nature Communications*.

REVIEWERS' COMMENTS

Reviewer #2 (Remarks to the Author):

The authors are to be commended on the additional experiments undertaken in response to the reviews of the initial manuscript. The examination of a cohort of FGF21-KO mice subjected to the overfeeding protocol clearly establishes the dispensability of FGF21 in the hypophagic response to experimental overfeeding. In addition, preliminary data evaluating the potential role of two proteins identified in the proteomic screen from overfed subjects clearly supports the importance of exploring putative secreted factors contributing to the physiologic response to forced overfeeding.

The revised manuscript has been significantly improved and will be of tremendous interest to those interested in the central mechanisms of energy balance control.

Reviewer #3 (Remarks to the Author):

The authors adequately answered reviewer 1's comments with additional experiments and analyses.